# Rainfall Potential and Consequences on Structural Soil Degradation of the Most Important Agricultural Region of Mexico



Mariano Norzagaray Campos [1,*], Patricia Muñoz Sevilla [2], Jorge Montiel Montoya [1], Omar Llanes Cárdenas [1], María Ladrón de Guevara Torres [3] and Luz Arcelia Serrano García [2]

1   Interdisciplinary Research Center for Regional Integral Development-IPN-Sinaloa Unit, National Polytechnic Institute, Blvd. Juan de Dios Bátiz Paredes No. 250. Colonia San Joachín, Guasave 81049, Mexico; jmontielm@ipn.mx (J.M.M.); ollanesc@ipn.mx (O.L.C.)
2   Interdisciplinary Research Center and Studies on Environment and Development (CIIEMAD), National Polytechnic Institute, Alcaldía Gustavo. A Madero, Mexico City 07340, Mexico; nmunozs@ipn.mx (P.M.S.); lugarcias@ipn.mx (L.A.S.G.)
3   Interdisciplinary Research Center for Regional Integral Development-IPN-Oaxaca Unit, National Polytechnic Institute Municipio de Santa Cruz Xoxocotlán, Oaxaca 71230, Mexico; maladron@ipn.mx
*   Correspondence: mnorzagarayc@ipn.mx; Tel.: +52-6871217072

**Abstract:** This study investigates the historical variability in annual average precipitation in the northwest region of Mexico, aiming to evaluate the cumulative impact of precipitation on soil degradation and associated risks posed by rainfall. Despite being known as "*The Agricultural Heart of Mexico*", the region's soil has experienced significant damage to its granulometric structure due to unpredictable rainfall patterns attributed to climate change. Sixteen historical series of average annual rainfall were analyzed as stationary stochastic processes for spectral analysis. The results revealed exponential decay curves in each radial spectrum, indicating a linear relationship between frequency and amplitude. These curves identified initial impulses correlated with moments of severity for structural damages caused by rainfall-induced degradation. The degradation process, exacerbated by water stress, accelerates, as evidenced by maps illustrating approximately 75% soil damage. In the context of climate change and the uncertainty surrounding soil responses to extreme meteorological events, understanding this phenomenon becomes crucial. Recognizing the dynamic nature of soil responses to environmental stressors is essential for effective soil management. Emphasizing the need to employ numerical processes tailored to new environmental considerations related to observed soil damages is crucial for sustainable soil management practices in any region.

**Keywords:** precipitation; soil degradation; climate variability; spectrum and soil management; Mexico

## 1. Introduction

Contemporary challenges facing Mexican agriculture, particularly in the northwest region, necessitate adaptation to meet evolving global market demands. The recent North American Free Trade Agreement offers renewed prospects, calling for enhanced commercial engagement and environmental management plan restructuring. However, the compounding challenges of climate change exacerbate pressure on agricultural practices, particularly soil degradation, presenting a complex task for adaptation amidst globalization [1,2].

Soil degradation in Mexico results from various granulometric combinations due to the distribution variation in edaphic richness across its territory. This diversity contributes to multiple factors affecting and transforming soil structure, leading to numerous methodologies and definitions to study its degradation [3]. However, existing results are solely close approximations of the actual level of soil degradation.

The differences in soil degradation across the Mexican Republic are evident in the literature [4]. One report indicates that 61.7% of the soil in the national territory is affected

by erosions caused by water, wind, chemicals, physical factors, and soil withering, leading to degradation [5] (pp. 45–47), [6] (p. 31). Meanwhile, another study, at a 1:250,000 scale, provides a map and reports similar causes of soil degradation but attributes it to anomalous management practices, particularly with agricultural cover areas, revealing a 55% total degradation trend that could lead to erosion [7].

Agricultural production is heavily impacted by soil degradation and changes in land use [8]. Soil degradation poses a significant challenge to agricultural productivity and economic performance, especially in the northwest region of Mexico. Despite variations in the degree of degradation observed in studies, adherence to current management practices poses a substantial risk to soil granulometric structures. This threat is further compounded by the ongoing role of agriculture in contributing to Global Greenhouse Gas emissions (*GHGs*). Recent studies indicate that agriculture, worldwide, is responsible for a significant portion of these emissions [8,9].

In terms of global agriculture's responsibility for its anthropogenic activities associated with food systems, encompassing both pre- and post-agricultural production, approximately 16.5 gigatons (*Gt*) of carbon dioxide ($CO_2$) equivalent are estimated to be emitted into the atmosphere annually. This constitutes around 30.55% of total emissions, roughly equivalent to one-third of the total emitted, which amounts to 54 $GtCO_2e$ year$^{-1}$ [9].

The continuous and rapid growth of the urban population will produce diminutions in arable land [10–13]. Fundamental changes in food production becomes evident. Given that agricultural activity will continue to play a significant role in *GHGs* emissions it is crucial to seek a shift towards sustainable practices to meet the increasing demands of a globalized world [14]. The absence of changes and functional methodologies to prevent soil degradation, closely linked to the agricultural production phases on farms and fields, will result in significant *GHGs* emissions, further exacerbated by the incidence of rainfall. Therefore, rigorous and urgent strategies are required to reduce emissions through improvements in crop production. This involves not only soil conservation-based practices but also specific methodological approaches in farm and field environments to confer environmental value to agricultural products in the face of *GHGs* [15,16]. These strategies should not only reduce emissions but also enhance overall sustainability, ensuring the well-being of populations globally [1,2].

The degradation of soil is intricately linked to the "*perfect cycle of agricultural production*", encompassing various emission phases from field activities to food processing and waste management [17,18]. Balancing food demands, mitigating environmental impacts on soil, and guaranteeing population well-being is complex yet crucial for creating a resilient and sustainable agricultural system, especially in regions like the "*Agricultural Heart of Mexico*" given the consistent year-round nature of its agricultural activities.

Therefore, the primary focus should be on identifying the severity of consequences resulting from the impact of meteorological phenomena on soil, particularly those resulting from accumulated historical rainfall over time ($\overline{P}_T(i,t)_j$) and causing rapid damage. For this purpose, a dataset spanning 51 years was utilized, consisting of 16 temporal signals of average annual precipitation ($\overline{P}(i,t)_j$), collected from meteorological stations operated by CONAGUA (1961–2011). A stochastic approach was employed to assess the severity of consequences and risks associated with soil degradation. The analysis was conducted in the frequency domain to observe variations in the power spectrum for each temporal signal. Additionally, a degree of radial integration was utilized to estimate and visually represent the radial spectral potential for each respective temporal signal.

The analysis considers the region's ongoing historical exploitation, its high susceptibility to desertification processes, and the influence of North American monsoonal dynamics. It begins with the premise that relying on rigorously controlled and verified data is essential for developing effective agricultural management plans that address climate change, emphasizing the paramount importance of data quality. Ensuring data quality is accomplished through the analysis of seasonal stochastic conditions, enabling the estimation of spectral

potential and the regionalization of soil erosion risk, serving as a crucial indicator of the soil granulometric state in relation to the cumulative risk posed by precipitation.

This work responds to the current need for a comprehensive management approach to soil conservation in Mexico. New plans or adjustments are necessary to effectively control, counteract, and combat soil erosion, particularly in light of the evolving conditions brought about by climate change [16]. These plans must be functional, tailored to specific areas, and based on long-term characterizations of soil behavior. The importance of addressing soil degradation in agricultural areas cannot be overstated, especially given the increasing demands on agricultural land and the challenges posed by climate change. Singular attention must be paid to these issues to ensure the sustainability of agricultural practices and the long-term viability of agricultural production in Mexico. Therefore, understanding the consequences of soil degradation induced by meteorological factors is crucial for developing effective soil management strategies.

## 2. Materials and Methods

### 2.1. Study Area

In the extensive coastal plain of Northwest Mexico lies the territory State of Sinaloa, which has two interconnected agricultural valleys nestled within the hydrographic basins of the "*Río Fuerte*" and "*Río Sinaloa*". Renowned for their high productivity, these valleys serve as vital arteries in a region where surface waters converge into the rivers of the same name as their respective basins. Originating from the lofty peaks of the Sierra Madre Occidental (*SMO*), these rivers traverse the valleys, and theirs flow variables during the year fluctuate with respect to annual seasonally before culminating in the Sea of Cortés or Gulf of California (Figure 1).

### 2.2. Historical Significance and Challenges

The Sinaloa River Basin, known as "*The Agricultural Heart of Mexico*", boasts a long history of agriculture dating back to pre-colonial times. Indigenous tribes like the "*Cahitas*" and "*Pimas bajos*" cultivated various crops for sustenance and trade, establishing the region as a hub of agricultural activity. During the mid-20th century, the region emerged as a global leader in cotton exports, aligning with the principles of the "*Green Revolution*" in the 1960s [19]. However, contemporary challenges, exacerbated by climate change, now threaten the region's agricultural sustainability. To tackle these issues effectively, it is imperative to gain a comprehensive understanding of the risks posed by extreme weather events, particularly rainfall-induced soil degradation [19,20].

Therefore, the objective of this investigation is comprehending the severity of consequences produced by historical-annual-average-rainfall-induced soil structural internal degradation in a pivotal Mexican economic area known for its high agricultural production and commonly called the "*Agricultural Heart of Mexico*". Emphasizing the urgency, it underscores the need for innovative and sustainable agricultural management practices tailored to this vital agricultural soil [19].

### 2.3. Analysis of Rainfall-Induced Soil Degradation and Stationarity Condition

Focused on rainfall-induced soil degradation, this study meticulously evaluates the consequences on agricultural soil caused by $\overline{P}_T(ij)_t$, emphasizing degradation within the internal granulometric structure within the initial 7 to 30 min of rainfall. Understanding the dynamics of soil degradation during this critical period is vital due to the kinetic energy carried by raindrops, which can lead to significant alterations in the soil matrix. To comprehensively assess the impact of $\overline{P}(i,j)_t$ on soil, the $\overline{P}_T(ij)_t$ and its specific effects on soil properties were analysed. In order to ensure precision and accuracy in the analysis, temporally stable information was employed in the numerical process. This involved transforming the raw data obtained from weather stations into a signal composed of sines and cosines over time, with variations in amplitude and frequency [21,22].

Scheme 1 was meticulously designed step by step to illustrate the relationship between the numerical tools used, thereby delineating and separating the hierarchy of each process undertaken to process the dataset and obtain temporally stable signal information. The decision to present Scheme 1 first is strategic. It serves as a comprehensive roadmap, guiding readers through the methodology from start to finish. By presenting the graphical representation of ideas and numerical tools upfront, readers gain an immediate understanding of the research approach and the sequence in which the numerical tools will be utilized. This approach enhances clarity and facilitates comprehension, ensuring that readers can follow the methodology seamlessly. In the subsequent sections of the manuscript, each numerical tool comprising the methodology will be elaborated upon step by step. This sequential approach allows for a detailed exploration of each tool's role and significance within the research framework. Furthermore, it enables readers to grasp the rationale behind the selection and application of each numerical tool, thereby enhancing transparency and rigor in the methodology section.

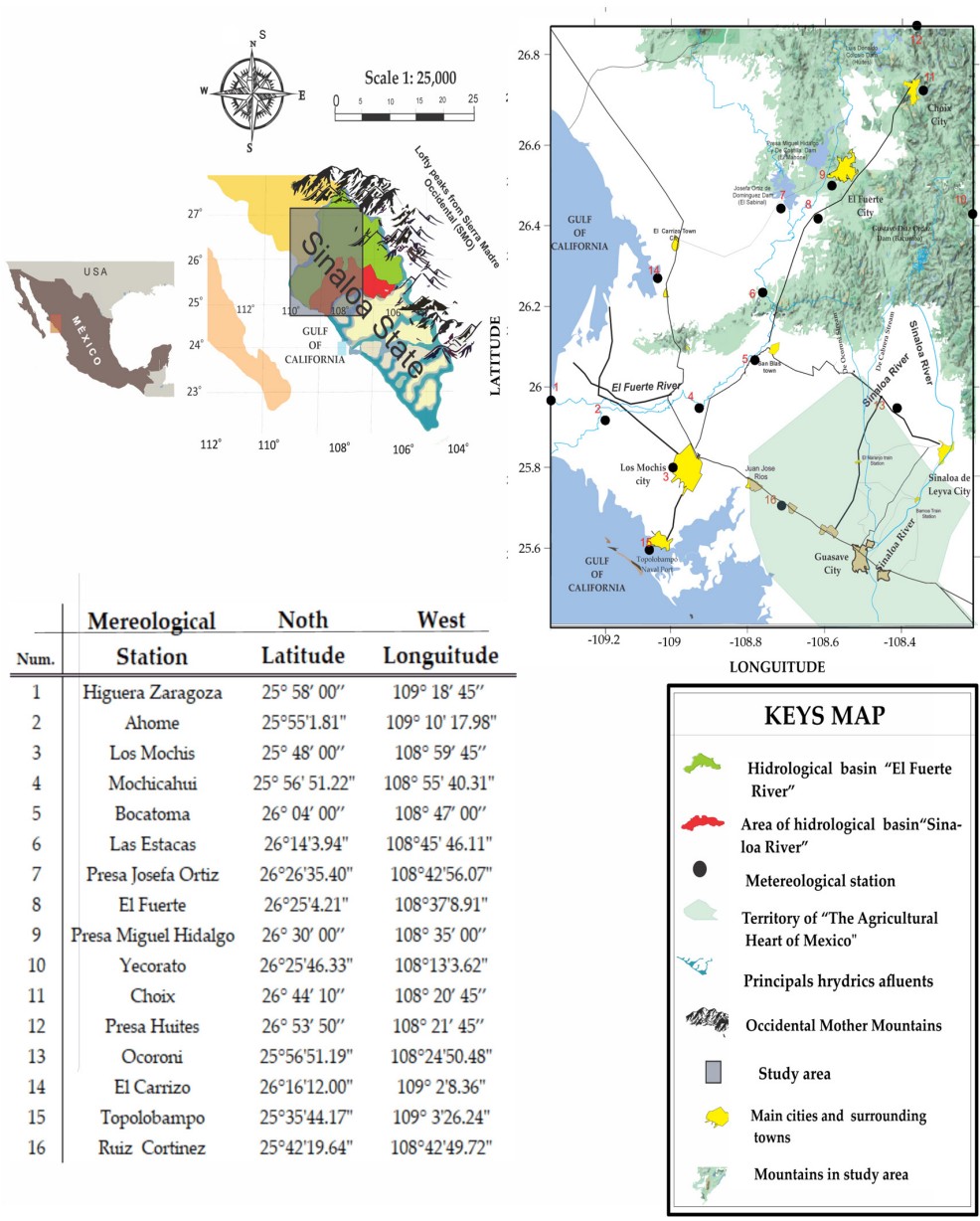

| Num. | Mereological Station | North Latitude | West Longuitude |
|---|---|---|---|
| 1 | Higuera Zaragoza | 25° 58' 00" | 109° 18' 45" |
| 2 | Ahome | 25°55'1.81" | 109° 10' 17.98" |
| 3 | Los Mochis | 25° 48' 00" | 108° 59' 45" |
| 4 | Mochicahui | 25° 56' 51.22" | 108° 55' 40.31" |
| 5 | Bocatoma | 26° 04' 00" | 108° 47' 00" |
| 6 | Las Estacas | 26°14'3.94" | 108°45' 46.11" |
| 7 | Presa Josefa Ortiz | 26°26'35.40" | 108°42'56.07" |
| 8 | El Fuerte | 26°25'4.21" | 108°37'8.91" |
| 9 | Presa Miguel Hidalgo | 26° 30' 00" | 108° 35' 00" |
| 10 | Yecorato | 26°25'46.33" | 108°13'3.62" |
| 11 | Choix | 26° 44' 10" | 108° 20' 45" |
| 12 | Presa Huites | 26° 53' 50" | 108° 21' 45" |
| 13 | Ocoroni | 25°56'51.19" | 108°24'50.48" |
| 14 | El Carrizo | 26°16'12.00" | 109° 2'8.36" |
| 15 | Topolobampo | 25°35'44.17" | 109° 3'26.24" |
| 16 | Ruiz Cortinez | 25°42'19.64" | 108°42'49.72" |

**Figure 1.** Geographical coordinates and map positions of weather stations in the coastal plain and lofty mountainous areas of Northwest Mexico are provided, with a focus on the productive soils of the "*River Sinaloa*" hydrographic basins, renowned as the "*Agricultural Heart of Mexico*" zone.

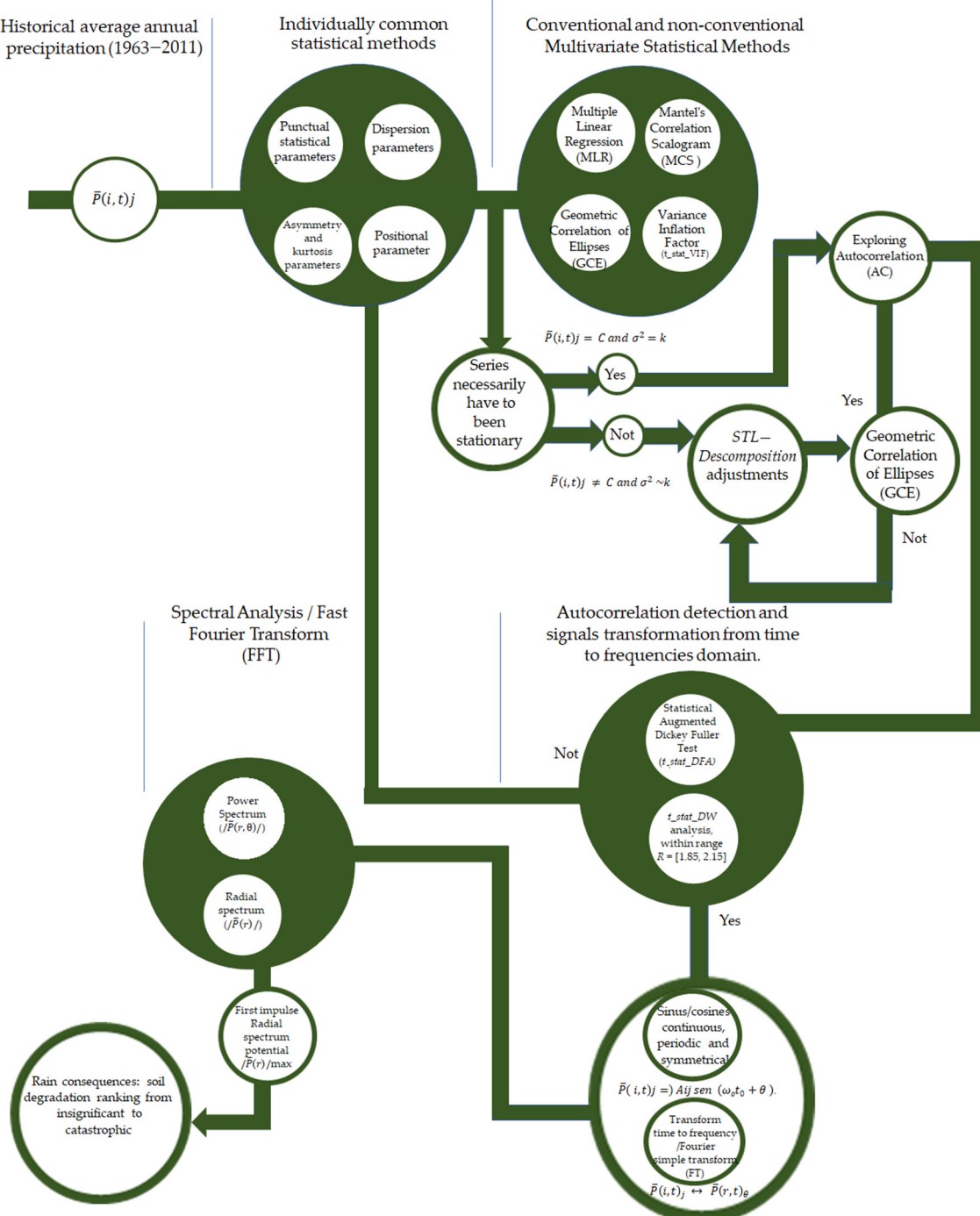

**Scheme 1.** In the clockwise direction, a sequential graphical representation of numerical tools and related concepts is shown, integrating the entire methodology utilized in this research.

*2.4. Assessment of Dataset Continuity*

To reduce dataset uncertainty, it was crucial to examine the errors in information collection methods at meteorological stations, including sensor data collection, human error, management, and processing, among others. These errors could introduce significant variability into the data, sometimes showing similarities in time-measured information and an exponential increase or decay in the *AC* function (*ACF*) over time. Therefore, each dataset underwent analysis for irregularities or *AC* using non-conventional statistical methods [23–25]. Considering that, initially, not all data series were stationary, it was essential to identify those meeting the continuity criteria before applying numerical processes to

obtain the function of sine and cosine. This involved analyzing data from 16 meteorological stations operated by CONAGUA to determine series meeting the continuity criteria.

### 2.5. Correlation Modelling and Randomness in the Dataset

Time-related measurements were investigated using a common multiple correlation modelling *RLM*. Physically traceable, non-conventional statistical methods were applied to identify potential *AC* and address dataset randomness. This approach aimed to enhance result confidence and prevent discrepancies in statistical hypothesis tests for seasonal conditions, thus avoiding erroneous stochastic model outcomes due to overlooked *AC* [26,27]. To analyze *AC* in the dataset and characterize the behavior of each signal, an individual examination of each dataset was conducted. After individual analysis, physically traceable, non-conventional statistical methods for determining *AC* were employed [27–29]. Graphical analysis of all datasets was conducted to detect discontinuities or deviations [30], ensuring that modelling simplifications adhered to established quality standards for describing seasonal statistical processes.

### 2.6. Individual Statistical Diagnoses: Conventional and Non-Conventional Multivariate Methods for AC Detection

The punctual statistical parameters used for individual diagnoses were the values' minimum ($\overline{P}(i, t)j_{min}$) and maximum ($\overline{P}(i, t)j_{max}$), media ($\overline{\overline{P}}(i,t)_j$), and median ($\overline{P}m(i, t)j$). The dispersion parameters comprised the absolute range ($R_{abs}$), standard deviation ($\sigma$), variance ($\sigma^2$), determination coefficient ($R^2$), correlation coefficient ($R$), and non-determination coefficient (($k = 1 - R^2$). The parameter to describe asymmetry $Swew\_\overline{P}_T(i, t)_j$, the kurtosis statistic $Kurt\_\overline{P}_T(i, t)_j$, and the e positional statistical parameters $Q25th$ and $Q75th$ percentiles were also used.

To detect *AC*, a combination of conventional and physically traceable, non-conventional statistical methods was employed. The conventional method utilized was Multiple Linear Regression (*MLR*), while the three non-conventional visual methods included Mantel's Correlation Scalogram (*MCS*), Geometric Correlation of Ellipses (*GCE*), and the Variance Inflation Factor (*t_stat_VIF*). Specifically, in the analysis with *t_stat_VIF*, the *AC* evaluation was based on the inflation factor, which is linked to the average variance [31,32]. The emphasis on using *t_stat_VIF* stemmed from its consistent and successful utilization across various works. It was employed to derive confidence intervals for site velocities derived from the Global Navigation Satellite System (*GNSS*), as well as from meteorology/climatology and soil contamination studies with multivariate data [33–35].

### 2.7. MLR Analysis

The absence of correlation indicated by *k* or the correlation indicated by $R^2$ obtained via *MLR* analysis as a first approximation were considered as partial indicators, because results focusing solely on one magnitude might overlook some degree of *AC*. This indirectly suggested that there was no *AC* when the variables were correlated or not with another *St*. This implied that if any degrees of *AC* were present in the dataset analyzed, they could be embedded within the percentage of *k* or $R^2$, making it challenging to detect. Therefore, it was recognized that specification errors might arise when trying to enforce the functionality of the model based solely on *k* or $R^2$ for *AC* usage [36]. The results from *MLR* for *AC* detection, due to disturbances and truncations in Minimum Ordinary Square (*MCO*), risked errors in hypothesis testing and potentially led to incorrect *AC* acceptance or rejection decisions. Hence, its results served as partial and preliminary indicators, as these disturbances could obscure the representation of a stationary stochastic process and affect *AC* determination [37]. Despite the ability to intuit *AC* through Fisher statistical probability contrasts (*prob_F_stat*) with a significance level $\alpha = 0.05$, the disturbances constrained *AC* analysis. To address these challenges and achieve more accurate *AC* identification, visual methods were employed [38]. These methods detected correlations and measured *ACF* by comparing correlations within and between the original dataset or each *St*.

*2.8. Mantel's Correlation Scalogram (MCS) Analysis*

The MCS analysis, the second method utilized, visually examined shared information within and between each *St*, enabling the configuration of a correlation regarding the Ahome dataset based on the maximum percentage spatial distribution that $R^2$ could acquire ($R^2 = 1 = 100\%$). Its aim was to detect anomalies within the datasets, focusing on both correlation and *AC*. The anomalies were visually distributed on a triangle's base for *AC* and below for correlation, effectively representing dataset characteristics. Simple correlation was presented in Mean Squared Coherence (*MCS*) as a percentage distribution of $R^2$ values, offering nuanced insights into shared variance. The graphical representation was based on a phased copy of the data series, expressed as a regression equation, illustrating the consecutive similarity between nominal values each time they were compared, as well as their original representation.

*2.9. Geometric Correlation of Ellipses (GCE) Analysis*

The *GCE* analysis employed a geometric approach to detect correlation and *AC* within the datasets. By forming a triangle using geometric figures, mainly trending towards ellipses or circles, this method offered insights into the dataset's variance. A trend towards ellipses indicated correlation within the triangle, while the presence of *AC* was defined by geometric figures tending towards an ellipse at the base of the triangle. The qualitative nature of the *GCE* method facilitated visual identification of *AC* by highlighting ellipses based on the number of signals, thereby indicating its presence.

*2.10. Variance Inflation Factor (t_stat_VIF) Analysis*

The *t_stat_VIF* analysis functioned as a quantitative method to assess multicollinearity within the datasets and its potential impact on the results obtained from *MLR*. By calculating the Variance Inflation Factor, this analysis offered insights into the degree of self-correlation present in each dataset. Values of *t_stat_VIF* > 10 indicated a significant self-correlation problem, suggesting *AC* presence. Conversely, values of *t_stat_VIF* < 10 implied the absence of self-correlation, thereby confirming the quality of the dataset for further analysis.

*2.11. Elimination of AC and Seasonality Assessment: Augmented Dickey–Fuller Test (t_stat_DFA) Analysis and Durbin–Watson Test (t_stat_DW)*

After identifying datasets with *AC*, a meticulous elimination process was initiated to remove randomness or noise, aiming to establish consistent seasonal stochastic processes over time by minimizing potential pre-existing variations. This involved maintaining uniformity in the behaviors of measured nominal values, preparing the datasets for subsequent analysis, and interpretation. Following the *AC* elimination process, comprehensive verification was conducted to ensure that internal variance was not shared within the datasets. This included additional analyses and adjustments, such as employing the Seasonal and Trend Decomposition Using Loess (*STL-Decomposition*) method, to confirm the absence of *AC* and validate the dataset's quality for further analysis.

Once *AC* was eliminated, statistical criteria were employed to assess seasonality within the datasets. Autoregressive analyses were conducted to ascertain the presence or absence of stationarity in the datasets derived from each original signal (*St*). The representation of the autoregressive characteristics within each dataset was expressed through the following equation:

$$a_1 \overline{P}i_{t-1} + a_2 \overline{P}i_{t-2} + \cdots + a_p \overline{P}i_{t-p} + \varepsilon_t. \tag{1}$$

Here, $a_1, a_2, \ldots\ldots, a_p$ represents the constant drift incorporated within $\overline{P}(i, j)_t$ and $\varepsilon_t$ denoted the historical white noise within each representation of *St*.

When $\overline{P}(i, t)j = \overline{P}(i, 0)j = 0$, in the absence of autocorrelation, Equation (1) can be reformulated as the following characteristic equation:

$$m^p - m^{p-1}a_1 - m^{p-2}a_2 - \cdots - a_p = 0. \tag{2}$$

Here, $m^p, m^{p-1}, m^{p-2}, \ldots, a_p$ represents the autocorrelation inherent in each *St* and at the initial moments of rainfall, represented by $\overline{P}(i,t)j = \overline{P}(i,0)j = 0$, it demonstrates a tendency toward unity. This result is equivalent to the presence of the unitary square ($I(1)$) of Equation (2). The behavior described by Equation (2), trending towards $I(1)$, implies non-stationarity in the stochastic process, and its statistical average parameters were defined as follows:

$$\overline{P}(i,t)j \neq C \text{ and } \sigma^2 \sim k \tag{3}$$

where $C$ and $k$ are constants indicators representing the absence of an average value for $\overline{P}(i,t)j$ and an undefined variance, respectively, for a non-stationary *St*.

The presence of $I(1)$ in the characteristic equation and its connection with non-seasonality or absence with the stationarity were examined to define the stationarity condition of each St. The *t_stat_DFA* analysis served as a final confirmation of the presence or absence of $I(1)$ within each dataset, with a critical level contrast of 3.5%. To safeguard the integrity of the $R^2$ value obtained from the *RLM* tests and to mitigate the risk of spurious results within the *t_stat_DFA*, precautions were taken to filter out any potential undetected AC. This involved conducting exploratory analyses on first differences (*PED*) or second differences (*SED*) to ascertain whether the datasets exhibited characteristics associated with non-stationarity. To validate the *t_stat_DFA* results and ensure the absence of *AC*, a prior *AC* contrast was conducted using the Durbin–Watson test (*t_stat_DW*).

### 2.12. Outlier Correction and Wave Representation for Spectral Analysis

Statistically identified outlies within the datasets were meticulously corrected to maintain temporal and frequency domain consistency. The adjusted information underwent spectral analysis following established criteria. The information must exhibit stationary behavior to facilitate a proper understanding of the source and the properties that generate $/\overline{P}(r, t)_\theta/$ and $/\overline{P}(r)/$. Therefore, using the stationary information, a deterministic wave representation of each dataset was established, allowing for numerical adjustments using Fourier series [39]. This ensured the reduction of uncertainty in the approximation of nominal values and facilitated the transformation from the time domain to the frequency domain. The *Fast Fourier Transformation* (*FFT*) was used to establish guarantees in the result obtained in the directions $r$ and $\theta$ to the spectrum of $/\overline{P}(r, t)_\theta/$ of each *St* on a regular mesh $m \times n = 15 \times 10$ with 150 finite elements. $/\overline{P}(r)/$ was graphed in a one-dimensional way to obtain a unique spectrum in each element of the mesh and visualize the set of low, medium, and high frequencies. These charts, based on spatial frequency or wave number, were considered as the natural logarithm of $/\overline{P}(r, t)_\theta/$.

### 2.13. Interpolation and Consequence Severity Scale

Limited data posed challenges in fully characterizing the $\overline{P}_T(i, t)_j$ variation, prompting interpolation between neighboring stations. The Kriging method [40] was employed, supported by evidence favoring geostatistical techniques for better estimates [41–43].

The limited data of $\overline{P}_T(i, t)_j$ posed challenges in fully characterizing its actual variation, leading to the need for an efficient configuration of the true spatial distribution of this historical accumulation with robust data interpolation. To achieve this, differences in the percentage of explained variability and root mean square error found in cross-validation of five types of Kriging interpolation techniques (ordinary, universal, with external drift, with individual variograms, and with combined variograms) were evaluated [40]. Ultimately, the interpolation using combined variograms was selected, as it provided comparable performance when applied to individual magnitudes of $\overline{P}_T(i, t)_j$. This demonstrated, consistent with other research, that Kriging interpolation with combined variograms could be successfully applied for real-time operations in the study area [41–43]. This approach proved crucial for understanding metrics like $/\overline{P}(r, t)_\theta/$ and $/\overline{P}(r)/$, revealing insights into the behavior of rain intensity curves, including their exponential decay concerning frequencies [44]. Results underwent rigorous evaluation against criteria specified by [45] (pp. 59–65), ensuring adherence to stringent spatial distribution standards. No-

tably, a 25 km radius was deemed suitable for flat coastal areas, while a 12 km radius was considered appropriate for mountainous regions of *SMO*.

Additionally, a severity scale was devised to evaluate the consequences of identified soil degradation risks, ranging from insignificant to catastrophic. This scale provided a framework for understanding potential impacts and guiding appropriate management strategies:

Insignificant: Risks may lead to minimal consequences. Implementing integrated conservation plans is recommended as a preventive measure.

Minor: consequences can be effectively managed through adaptive plans addressing climate change, with a focus on existing risks in intensive agriculture.

Moderate: risks demand substantial time and effort for mitigation.

Important: risks carry significant short-term consequences, necessitating detailed soil studies.

Catastrophic: risks present severe challenges for soils unsuitable for sustainable agriculture.

### *2.14. Data Compilation and Software Utilization*

The compiled information was organized into tables using *EXCEL 12.0* software. Various analytical methods, including statistical parameters; *RLM* estimator; the *GCE* method; *MCS*; *t_stat_VIF*; *STL-Decomposition* adjustments; and significance testing *t_stat_DW*, *t_stat_DFA*, and *Prob(F-statistic)* with respect to $\alpha = 0.05$ were conducted using *XVIEW 12.0* software. Additional analyses such as *MSC* and *spectral analysis* were performed using the PAST 5.0 program, while interpolations and the verification of results were executed using SURFER 10.0. Finally, the refinement of maps and figures was achieved using COREL DRAW 2018.

### 3. Results

The graphs in part A of Figure 2 show the behaviors of the nominal values of the matrix $m \times n = 15 \times 51$, comprising 761 elements designed to analyze the 16 *St* of the $\overline{P}(i, t)j$ original. Part B of Figure 2 displays the accumulated/annual $\overline{P}(i, t)j$ in the 16 meteorological stations. Concerning the total $\overline{P}(i, t)$ for the period 1961–2011 in each season, a concentration of 38% was observed in the 1980s. In part B, it is also evident that, within the study period, the stations of Yecorato and Mochicahui, respectively, exhibited the highest and lowest accumulation of $\overline{P}(i, t)j$, measuring 811.22 mm and 260.99 mm.

Parallel to the Sea of Cortez are the meteorological stations of the coastal plain, characterized by spatial variability in $\overline{P}(i, t)j$, with lower average values at $\overline{P}(i, t)j = 346.32$ mm. Conversely, stations located near the Sierra Madre Occidental (SMO) exhibit a higher average, with a cumulative value for $\overline{P}(i, t)j = 612.90$ mm. The station with the lowest rainfall, noted at $\overline{P}(i, t)j = 408.45$ mm, is represented by data from the Ruiz Cortines station, while the smallest magnitude, with data configured for the aforementioned Mochicahui station, measures $\overline{P}(i, t)j = 260.99$ mm.

Within *SMO*, the station with the greatest magnitude corresponds to Yecorato, while the smallest, measuring 434.78 mm, is Las Estacas. Similarly, in the vicinity of *SMO*, the three largest catchments in mm were recorded, with an average total of $\overline{P}(i, t)j = 811.22$, 809.45, and 737.28, respectively, for the stations of Yecorato, Huites, and Choix. Conversely, stations with smaller magnitudes, measuring $\overline{P}(i, t)j = 260.99$, 311.68, and 371.421 mm, are identified, respectively, at the stations of Mochicahui, Higuera de Zaragoza, and El Carrizo in the coastal plain.

The graph in part A of Figure 2 represents the temporal variation in the original 16 *St*, constructed to conduct one-dimensional analysis on the behavior of $\overline{P}(i, t)j$. This analysis revealed irregularities; for instance, in 1986, an anomalous magnitude of $\overline{P}(i, t)j = 258.59$ mm, the lowest annual average within all 765 nominal values ($m = 15$, $n = 51$), was observed. The behavior continued with high annual values characterized across the 1980s: 1980 (641.11 mm), 1982 (640.24 mm), 1983 (760.42 mm), and 1989 (639.20 mm). The 1990s showed an irregular trend, with biannual minimum values of $\overline{P}(i, t)j$ as follows: 1994 (345.42 mm), 1996 (344.38 mm), and 1998 (364.20 mm). In 2003, an isolated and sporadic magnitude of $\overline{P}(i, t)j = 671.30$ mm was observed.

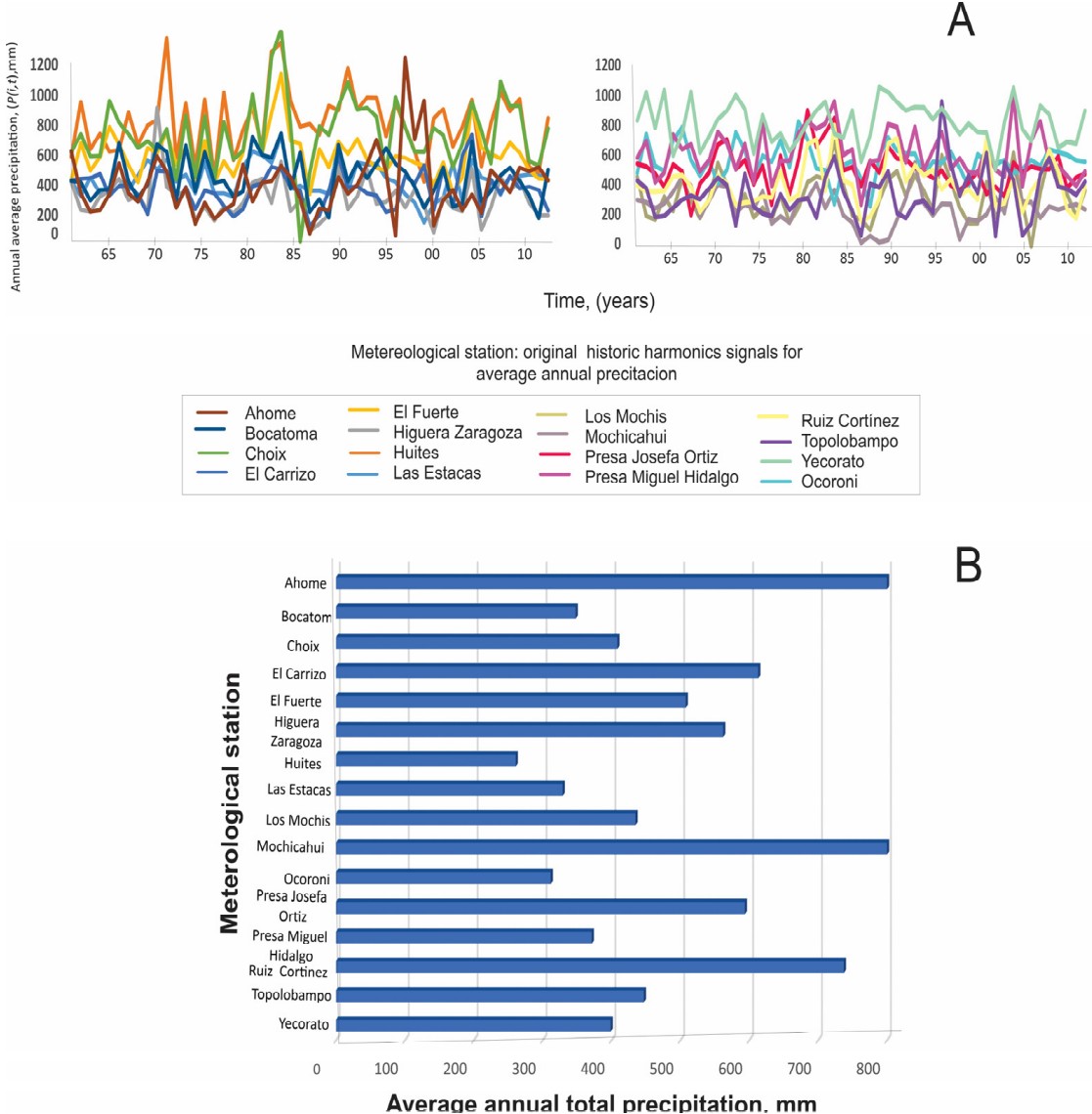

**Figure 2.** Configurations from 1961 to 2011: (**A**) Historical average annual precipitation (mm) for 16 meteorological stations in the "*Río Fuerte*" and "*Rio Sinaloa*" basins. (**B**) Annual accumulation (mm) at the same stations.

To classify the anomaly of 2003, considering the range $R = [760.42, 288.50]$ corresponding to the maximum and minimum accumulated annual variation of $\overline{P}(i,\ t)j$ in the area, it was deemed a medium to high rainfall event. This year marked the highest rainfall in the first decade of the twentieth century. If this outlier is disregarded, magnitudes fall within the range $R = [376.90, 431.16]$, revolving around the value $\overline{P}(i,\ t)j = 495.89$ mm. The behavior of the first decade of 20th century presents cyclical behaviors of maximums and minimums: 2001 ($\overline{P}(i,\ t)j = (373.90$ mm)), 2002 ($\overline{P}(i,\ t)j = 440.18$ mm), 2004 ($\overline{P}(i,\ t)j = 389.14$ mm), 2009 ($\overline{P}(i,\ t)j = 392.19$ mm), and 2010 ($\overline{P}(i,\ t)j = 431.16$ mm).

Table 1 presents the results of individual statistical diagnosis applied to the original 16 datasets. The indictor punctual parameters of central tendency present a range variation for $\overline{P}(i,\ t)j_{min}$ from 10.01 to 506.90 mm and for $\overline{P}(i,\ t)j_{max}$ from 1417.60 to 551.50 mm; they also occur for $\overline{\overline{P}}(i,t)_j$ from 811.22 to 261.00 mm and for $\overline{P}m(i,\ t)j$ from 8816.10 to 264.18 mm. The statistical dispersion parameters show $R_{abs}$ values from 1417.60 to 551.50, σ values from 228.42 to 109.37, $σ^2$ values from 52,174.74 to 11,962.57, a *CV* range from 51.53 to 16.62, $R^2$ values from 0.11 to $2 \times 10^{-3}$, and *R* values from 0.17 to 0.52. The variation in

asymmetry and shoring statistical parameters, respectively, present the following magnitudes: $Skew\_\overline{P}_T(i, t)_j$ from 1.63 to $-0.16$ and $Kurt\_\overline{P}_T(i, t)_j$ from 5.53 to $-0.90$. Finally, the positional statistical parameter for percentile 25% values ranges from 714.80 to 193.90 and for percentile 75% values from 945.01 to 314.30.

Concerning Ahome meteorological station, there is individual variation of $R = 0.45$ in shared information with the other 15 *Sts*, suggesting high correlation in the information, with all *Sts* displaying undefined trends in positive and negative moments (Figure 2). This high correlation suggests a potential *AC* within the *St*, which could impact the spectral analysis of $\overline{P}(i, t)j$ via signals from seasonal statistical processes.

For spectral analysis, it was considered preferable to work with seasonal stochastic processes due to their simpler and more stable statistical properties compared to non-seasonal processes. Non-seasonal processes can be more complex, as they may accumulate *AC* over time, compromising the results and introducing uncertainty and deviation from reality if not considered during the verification of seasonality assumptions.

The evolution of nominal values over time in a multiple analysis of $\overline{P}(i, t)j$ for all meteorological stations exhibits a consistent behavior around the constant level $\overline{P}(i, t)j = 504.53$, with permanent changes dispersed in a joint variability, as indicated by the values $\sigma = 134.36$ and $\sigma^2 = 18,052.72$. The result $R^2 = 0.21$ with respect to Ahome meteorological station suggests that they share 21% of their information. The distribution of nominal values more or less converges at the same point measure of $\overline{P}(i, t)j = 487.71$. Their asymmetry, defined by $Skew\_\overline{P}(i, t)j = 0.453$, indicates a slight positive concentration of nominee values, which are slightly elongated to the right, while $Kurt\_\overline{P}_T(i, t)_j = 2.17$ indicates a curve characterized due to normal or curved behavior.

The individual diagnosis and multiple analysis statistics of the *St* indicate a high level of shared information, both suggesting a potential *AC* within the *St* that may impact the spectral analysis of $\overline{P}_T(i, t)_j$ due to signals from non-seasonal statistical processes.

Given that working with seasonal stochastic processes is more straightforward due to their simpler and more stable statistical properties compared to non-seasonal processes, which can be complex and varied, and considering the possible *AC* within the time series that could invalidate results, leading to uncertainty and deviation from reality, it became imperative to eradicate the level of *AC* in each *St* to verify the assumptions of seasonality in the information.

The following are the results of four statistical techniques used to analyze *AC* conditions. These results correspond to respective algorithms detecting *AC* within the *Sts*. It is important to remember that the presence of *AC* would invalidate any seasonality condition. The results of the first statistical tool to detect *AC* are presented in Table 2 and they correspond to a low shared variance response of Ahome station compared to the rest of the *Sts*, making this station the dependent variable. The $R^2$ values range from 0 to 1, suggesting 100% similarity of information. A 21% correlation with $R^2$ was considered possible with *AC*.

Contrasts [*prob. (F_stat)* vs. $\alpha$], with values for *prob. (F_statistc)* in the range [0.8, 0.33], exceeded $\alpha < 0.05$, indicating no significant differences between nominal values of $\overline{P}(i, t)j$ in almost all 16 *Sts*. *Prob.* $> 0.05$ in the contrasts [prob. (*prob. (F_stat)* vs. $\alpha$] was associated with stations presenting the greatest strength in the absence of correlation (greater difference in shared information). The stations that exhibited the greatest strength in the absence of correlation (indicating greater difference in shared information) include Bocatoma (*prob.* = 0.74), Choix (*prob.* = 0.60), El Fuerte (*prob.* = 0.84), Las Estacas (*prob.* = 0.71), Ocoroni (*prob.* = 0.81), Presa Josefa Ortiz (prob. = 0.79), and Yecorato (*prob.* = 0.80).

**Table 1.** Statistical analysis of 16 time series (1961–2011) for estimated central tendency, dispersion, asymmetry, and statistical bolstering in average annual precipitation in northwestern Mexico's mountainous zones and coastal plains.

| $\overline{P}(i,t)_j$ | Higuera Zaragoza | Ahome | Los Mochis | Mochicahui | Bocatoma | Las Estacas | Presa Josefa Ortiz | El Fuerte | Presa Miguel Hgo. | Yecorato | Choix | Huites | Ocoroni | El Carrizo | Topolobampo | Ruiz Cortínez |
|---|---|---|---|---|---|---|---|---|---|---|---|---|---|---|---|---|
| $\overline{P}(i,t)_{jmin}$ | 87.10 | 68.20 | **10.00** | 26.60 | 180.00 | 152.00 | 209.60 | 338.20 | 276.80 | **506.90** | 27.40 | 418.50 | 264.50 | 127.30 | 69.50 | 164.00 |
| $\overline{P}(i,t)_{jmax}$ | 905.80 | 1228.40 | 683.70 | **551.50** | 737.10 | 724.20 | 895.20 | 1122.00 | 993.20 | 1042.50 | **1417.60** | 1357.40 | 821.00 | 729.50 | 953.00 | 797.90 |
| $\overline{\overline{P}}(i,t)_j$ | 311.69 | 398.66 | 329.27 | **261.00** | 446.91 | 434.79 | 507.84 | 593.56 | 613.00 | **811.22** | 737.29 | 809.46 | 562.10 | 371.42 | 347.89 | 408.45 |
| $\overline{P}m(i,t)_j$ | 282.30 | 365.00 | 328.80 | **264.18** | 421.60 | 431.99 | 500.76 | 581.50 | 580.10 | **816.10** | 733.70 | 805.43 | 556.57 | 383.10 | 341.12 | 375.60 |
| $R_{abs}$ | 905.80 | 1228.40 | 683.70 | **551.50** | 737.10 | 724.20 | 895.20 | 1122.00 | 993.20 | 1042.50 | **1417.60** | 1357.40 | 821.00 | 729.50 | 953.00 | 797.90 |
| $\sigma$ | 138.83 | 205.44 | 144.59 | 112.78 | 147.77 | 109.99 | 127.54 | 153.47 | 153.21 | 134.84 | **228.42** | 213.40 | **109.37** | 120.08 | 159.95 | 147.22 |
| $\sigma^2$ | 19,273.46 | 42,205.03 | 20,906.51 | 12,719.91 | 21,835.74 | 12,097.66 | 16,265.98 | 23,553.16 | 23,472.22 | 18,181.87 | 52,174.74 | 45,537.87 | **11,962.57** | 14,418.70 | 25,585.30 | 21,673.28 |
| $CV$ | 818.70 | 1296.60 | 683.70 | **578.10** | 917.10 | 876.20 | 1104.80 | 1460.20 | 1270.00 | 1549.40 | 1445.00 | **1775.90** | 1085.50 | 856.80 | 1022.50 | 961.90 |
| $R$ | 0.09 | 1.00 | 0.04 | 0.04 | 0.03 | 0.001 | 0.01 | 0.01 | 0.01 | 0.002 | 0.01 | **0.0002** | 0.00 | **0.11** | 0.03 | 0.05 |
| $R^2$ | 0.45 | **0.52** | 0.44 | 0.43 | 0.33 | 0.25 | 0.25 | 0.26 | 0.25 | **0.17** | 0.31 | 0.26 | 0.19 | 0.32 | 0.46 | 0.36 |
| $Skew\_(\overline{P}(i,t)_j)$ | **1.63** | 1.61 | 0.34 | 0.06 | 0.23 | 0.24 | 0.65 | 1.16 | 0.34 | −0.12 | 0.10 | 0.43 | **−0.16** | 0.34 | 0.96 | 0.85 |
| $Kurt\_(\overline{P}(i,t)_j)$ | **5.53** | 4.81 | **−0.24** | 0.51 | −0.90 | 0.23 | 1.60 | 2.29 | −0.26 | −0.68 | 1.90 | 0.29 | 1.10 | 0.41 | 2.83 | 0.38 |
| $Q25th$ | 227.50 | 248.30 | 198.00 | **193.90** | 332.00 | 344.70 | 413.30 | 493.90 | 493.80 | **714.80** | 590.50 | 633.50 | 502.60 | 255.80 | 223.60 | 304.40 |
| $Q75th$ | 367.80 | 496.27 | 430.20 | **314.30** | 576.80 | 531.90 | 568.80 | 669.20 | 746.00 | 908.70 | 894.10 | **945.00** | 619.81 | 443.87 | 438.40 | 469.10 |

$\overline{P}(i,t)_{jmin}$ and $\overline{P}(i,t)_{jmax}$ = maximum and minimum limits in mm, $\overline{\overline{P}}(i,t)_j$ = arithmetic mean, $\overline{P}m(i,t)_j$ = median, $R_{abs}$ = absolute range, $\sigma$ = standard deviation of the distribution, $\sigma^2$ = variance of the distribution, $CV$ = coefficient of variation, $R$ = coefficient of dispersion, $R^2$ = coefficient of correlation, $Skew\_(\overline{P}(i,t)_j)$ = coefficient of asymmetry, $Kurt\_(\overline{P}(i,t)_j)$ = coefficient of kurtosis, $Q25th$ = percentile 25th, and $Q75th$ = percentile $Q75th$.

**Table 2.** Multiple Linear Regression (*MLR*) of 16 historical series concerning shared variance in average annual precipitation with Ahome station on the coastal plains and mountain zones of northwestern Mexico.

| Multiple Linear Regression (*RLM*)/Dependent Variable: AHOME Method: Least Squares | | | | |
|---|---|---|---|---|
| **Sample: 1961–2011** | | **Included Observations: 51** | | |
| **Variable** | **Coefficient** | **Std. Error** | **t-Statistic** | **Prob.** |
| C | 281.08 | 283.59 | 0.99 | 0.33 |
| Ahome | 0.13 | 0.39 | 0.34 | 0.74 |
| Bocatoma | 0.13 | 0.24 | 0.53 | 0.60 |
| Choix | 0.46 | 0.38 | 1.23 | 0.23 |
| El Carrizo | −0.09 | 0.44 | −0.20 | 0.84 |
| El Fuerte | 0.35 | 0.41 | 0.85 | 0.40 |
| Higuera Zaragoza | −0.13 | 0.24 | −0.56 | 0.58 |
| Huites | −0.18 | 0.46 | −0.38 | 0.71 |
| Las Estacas | −0.30 | 0.49 | −0.62 | 0.54 |
| Los Mochis | 0.33 | 0.38 | 0.86 | 0.40 |
| Mochicahui | −0.01 | 0.30 | −0.04 | 0.97 |
| Presa Josefa Ortiz | 0.12 | 0.47 | 0.26 | 0.79 |
| Presa Miguel Hidalgo | −0.35 | 0.42 | −0.84 | 0.41 |
| Ruiz Cortínez | 0.23 | 0.36 | 0.65 | 0.52 |
| Topolobampo | 0.17 | 0.30 | 0.59 | 0.56 |
| Yecorato | −0.08 | 0.30 | −0.26 | 0.80 |
| R-squared | 0.21 | Mean dependent var. | | 398.65 |
| Adjusted R-squared | −0.12 | S.D. dependent var. | | 205.43 |
| S.E. of regression | 217.73 | Akaike info criterion | | 13.85 |
| Sum squared resid. | 1,659,201 | Schwarz criterion | | 14.46 |
| Log likelihood | −337.31 | Hannan–Quinn criteria | | 14.08 |
| F-statistic | 0.63 | Durbin–Watson stat (*t_stat_DW*) | | 1.71 |
| Prob (F-statistic) | 0.85 | | | |

The counterpart of $R^2$, namely $k^2 = 1 - R^2$ (coefficient of alienation or indeterminacy), demonstrated incidences between the nominal comparatives and the total proportion of $\sigma^2$, attributing 79% of the information to no correlation. This suggests the possibility of *AC* within any *St*. The *RLM* result of 21% of $R^2$ hints at potential disturbances within an *St*, hindering its representation as a stationary stochastic process. Consequently, the hypothesis $H_0$ was initially rejected, and the alternative $H_1$ was accepted.

Direct detection of *AC* using contrasts ([*prob. (F_statisitc)* vs. $\alpha$]) between the probabilities of *F_statisitc* and the significance level $\alpha = 0.05$ serves as a partial indicator of *AC*. However, the methodology's uncertainties regarding specification errors led to hesitation in assuming the absence of *AC*. The *RLM* method alone did not offer sufficient guarantees to conclusively argue against *AC* in the models analyzed.

Hence, three statistical tools were employed to visually identify areas where *AC* might occur. The second method examined the spatial distribution of variation in similarity, illustrating a 21% information concentration. It also identified *AC* by comparing signals along a linear distribution, visually represented at the scaleogram's base. While both the *RLM* and *MSC* use $R^2$ for analysis, the latter detects areas of *AC*, as depicted in Figure 3. The red vertex of the triangle highlights the high similarity between the first and last values of the matrix $m \times n$. Similarly, the base of the triangle in the same color indicates comparison pairs showing similarity.

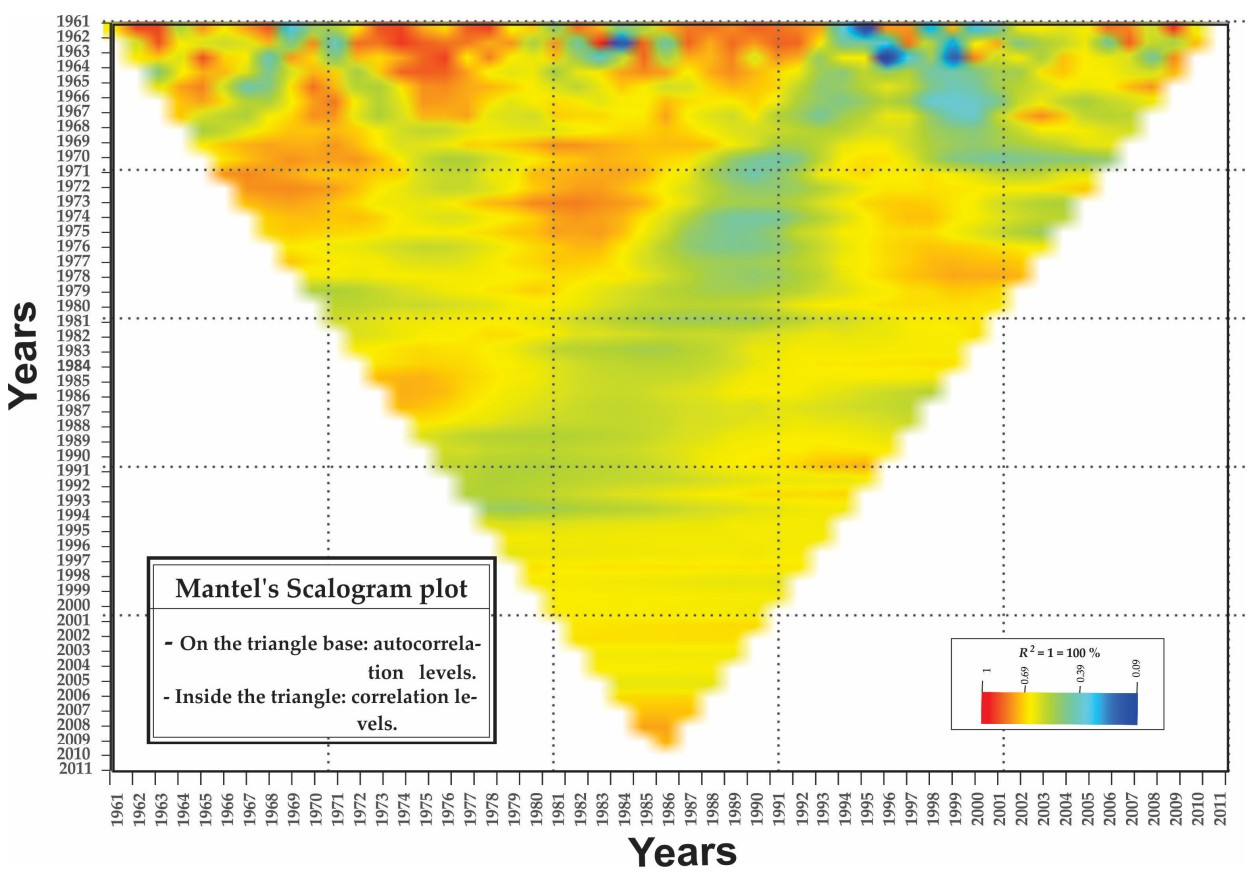

**Figure 3.** Mantell scalogram illustrating spatial distribution of average similarity among historical harmonics within the triangle. At the base, it compares similarity to a delayed copy of itself on each harmonic, showing correlation and autocorrelation levels from 0 to 1.

Although the *RLM* initially suggested an absence of *AC* (Table 1), the *MSC* reveals *AC* within the information previously assumed to be free of AC. This new insight offers a more comprehensive understanding, discouraging reliance solely on contrast tests for analyzing seasonality assumptions in the *St*.

To identify the *Sts* potentially affected by the outlier *AC* effect and to ensure accurate data representation with series of seasonal stochastic processes, the third method—*GCE*—was employed and the results are depicted in Figure 4. Figure 4 indicates stations that did not exhibit *AC* or correlation, meaning they did not share their $\sigma^2$ with another *St* or with themselves. Conversely, the figures in red within the triangle show stations displaying *AC* when their information is compared. Additionally, stations with no correlation when compared with each other are indicated within the triangle. Out of the 120 geometric figures formed as a result of the contrasts, 28 tended to form an ellipse, indicating 23.33% shared information. This result closely aligns with that of the *RLM* ($R^2 = 0.21$).

The contrasts of the 92 ellipses that tended to form circles instead of exact ellipses are exempt from correlation and *AC*. Specifically, the ellipses enclosed in a rectangle and marked in red in Figure 4 indicated *AC* when compared with themselves through this method. The tendency to form ellipses shows that the next eight datasets exhibit *AC*: Choix, El Fuerte, Higuera Zaragoza, Huites, Los Mochis, Mochicahui, Ruiz Cortines, and Topolobampo.

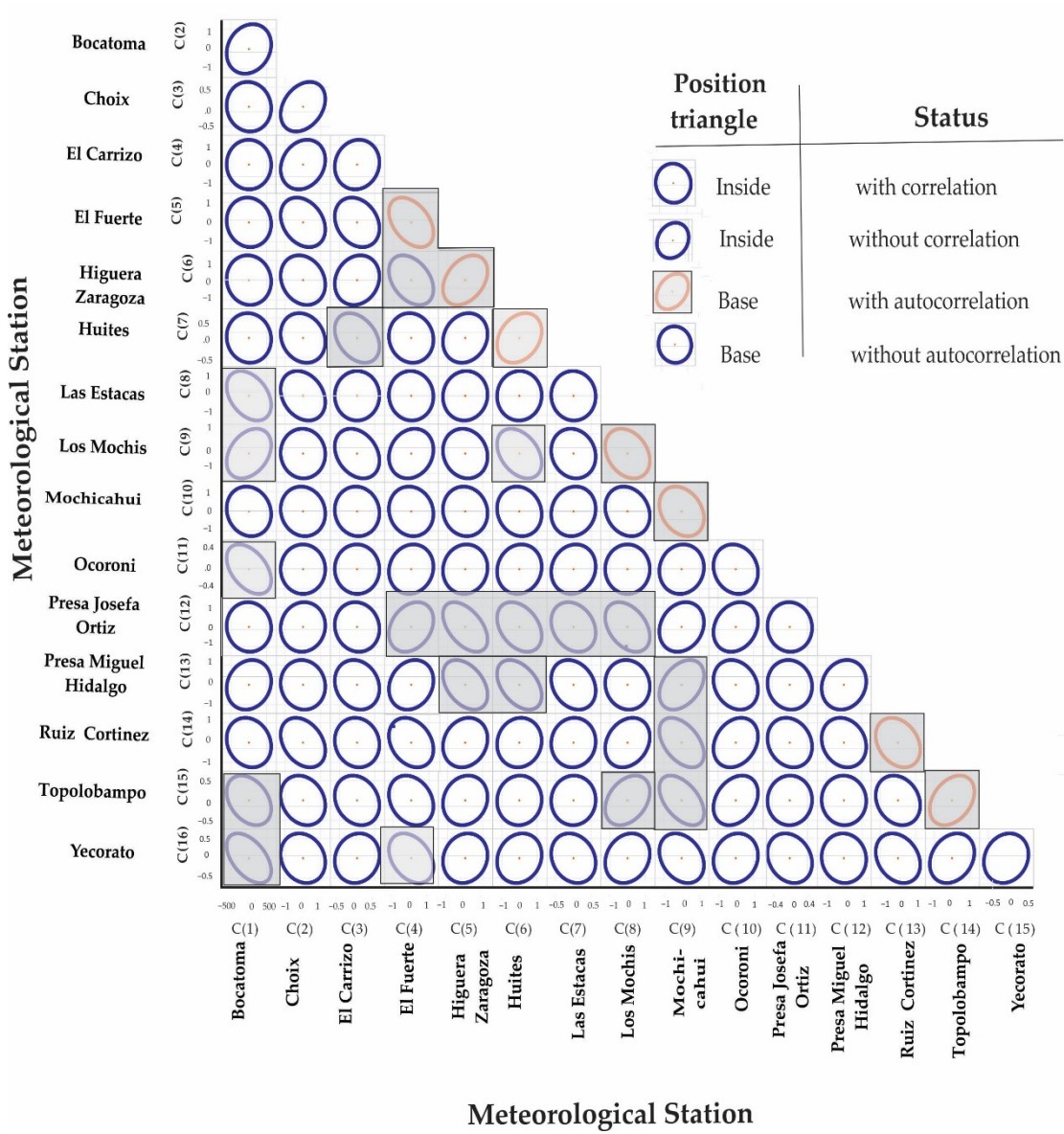

**Figure 4.** Algorithm correlating ellipses. The red geometric shapes at the triangle's base indicate a tendency for ellipses associated with harmonic autocorrelation.

The *GCE* analysis revealed that 50% of the Sts did not exhibit *AC*, while the remaining 50% required adjustment to achieve stationarity.

The random effects detected in *GCE* analysis in the dataset were found to deviate from the actual conditions observed in rainfall patterns. To address this, it was necessary to process the dataset of these eight *Sts* using the *STL-Decomposition* method.

To verify if the *AC* was reduced, the eight Sts underwent adjustment, and were retested using *GCE* and *t-stat-VIF* methods. Figure 5, part A, displays the graphical representation of the adjusted eight Sts with *STL-Decomposition*, while part B confirms reduced AC, as indicated by the change from ellipses tending to form circles. With this reduction in effect, a new *RLM* was conducted. The new results of $R^2 = 0.12$ and $R^2$ adjusted $= -0.26$, reported in Table 3, were considered acceptable, thereby increasing confidence in the reduction of AC within the *Sts*. This reduction allowed for the application of the *t_stat_DFA* contrast to verify the seasonality condition within the dataset.

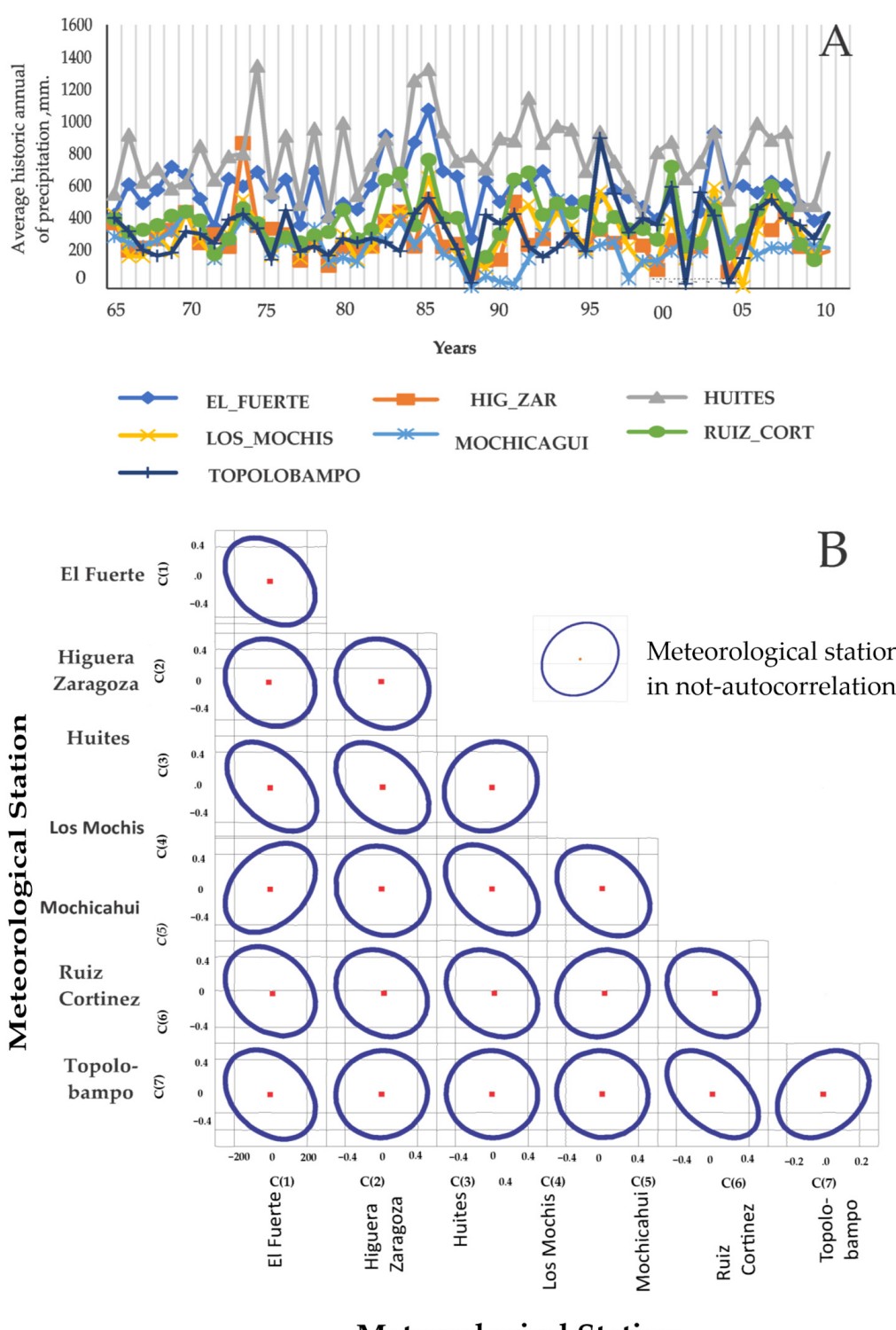

**Figure 5.** (**A**) Displays the graphical representation of the adjusted 8 *Sts* with *STL-Decomposition*. (**B**) Confirms the reduced *AC*, as indicated by the change from ellipses tending to form circles.

**Table 3.** Multiple Linear Regression was performed on 15 historical series from the Ocoroni station, exhibiting high initial correlation (Probability = 0.972). Subsequently, 50% of the non-stationary series were adjusted using *STL_Decomposition* to attain their respective stationary behavior.

| Multiple Linear Regression (*RLM*)/Dependent Variable: AHOME Method: Least Squares | | | | |
|---|---|---|---|---|
| Sample: 1961–2011 | | Included Observations: 51 | | |
| Variable | Coefficient | Std. Error | t-Statistic | Prob. |
| C | 428.81 | 144.62 | 2.97 | 0.01 |
| Ahome | 0.00 | 0.10 | 0.00 | 1.00 |
| Bocatoma | −0.02 | 0.22 | −0.07 | 0.95 |
| Choix | −0.01 | 0.14 | −0.05 | 0.96 |
| El Carrizo | −0.14 | 0.21 | −0.66 | 0.52 |
| El Fuerte | −0.09 | 0.24 | −0.36 | 0.72 |
| Higuera Zaragoza | −0.14 | 0.24 | −0.58 | 0.57 |
| Huites | −0.08 | 0.13 | −0.61 | 0.54 |
| Las Estacas | 0.00 | 0.26 | 0.01 | 0.99 |
| Los Mochis | −0.12 | 0.29 | −0.40 | 0.69 |
| Mochicahui | 0.19 | 0.22 | 0.83 | 0.41 |
| Presa Josefa Ortiz | 0.08 | 0.26 | 0.31 | 0.76 |
| Presa Miguel Hidalgo | 0.09 | 0.24 | 0.37 | 0.72 |
| Ruiz Cortínez | 0.21 | 0.21 | 1.00 | 0.32 |
| Topolobampo | 0.00 | 0.17 | −0.02 | 0.98 |
| Yecorato | 0.20 | 0.16 | 1.23 | 0.23 |
| R-squared | 0.12 | Mean dependent var. | | 562.10 |
| Adjusted R-squared | −0.26 | S.D. dependent var. | | 109.37 |
| S.E. of regression | 122.67 | Akaike info criterion | | 12.70 |
| Sum squared resid. | 526,68 | Schwarz criterion | | 13.31 |
| Log likelihood | −308.05 | Hannan–Quinn criteria | | 12.93 |
| F-statistic | 0.32 | Durbin–Watson stat (*t_stat_DW*) | | 1.89 |
| Prob (F-statistic) | 0.99 | | | |

Note that the results in Tables 2 and 3 highlight differences between the results obtained for the *F_stat* and *Prob(F_stat)* statistics in the original 16 datasets and the new results after adjusting the eight *Sts* to achieve stationarity. These adjusted datasets were then reintroduced to obtain the second *RLM* shown in Table 3. The first *RLM* in Table 1 presents values of *F_stat* = 0.63, *Prob(F_stat)* = 0.82, and $R^2$ = 0.21. In contrast, the second *RLM* in Table 3 shows these magnitudes distributed as follows: *F_stat* = 0.34, *Prob(F_stat)* = 0.98, and $R^2$ = 0.11. It is evident that there was a reduction after adjusting for seasonality using the *STL-Decomposition* technique.

The results of *t_stat_FIV* are displayed in Table 4, where the third column facilitates the establishment of the contrast [*t_stat_FIV* vs. *VIF* < 10]. It is notable that all values in this column fall within the range R = [1.3, 5.71], indicating that the magnitudes of *t_stat_FIV* < 10. Consequently, these findings suggest a complete absence of heteroskedasticity and correlation, effectively dismissing the potential presence of *AC* in the representation of $\overline{P}(i, t)j$.

Applying this new condition to the dataset prepares the foundation for applying *t_stat_DFA* to ascertain the presence of seasonality in each *St*, employing a 5% level of critical values set at −3.5 for comparison with *t_stat_DFA*. This method offers full confidence in determining the presence or absence of seasonality, allowing us to either accept or reject the hypothesis "$H_0 = \overline{P}(i, t)j$ "has a unit root exogenous: constant and linear trend" and, depending on its presence or absence, decide whether the datasets of $\overline{P}(i, t)j$ are stationary or non-stationary.

The outliers have been identified and corrected. Consequently, the following section will present the results verifying seasonality in the *Sts* using *t_Stat_DFA* to accept or reject the Ho hypothesis regarding the presence or absence of *I*(1), associated, respectively, with seasonality or non-seasonality in any St. The results of the *t_Stat_DFA* contrasts are

displayed in Table 5. This diagnosis reflects the outcomes obtained after all 16 $\overline{P}(i,\ t)j$ series passed graphical inspection and demonstrated the absence of *AC*.

**Table 4.** Results of *t_stat_FIV* in 16 datasets of historical series, using the significance level *VIF* < 10 associated with the maximum value to search for multicollinearity in respective nominal factors.

| | **Variance Inflation Factors (*t_stat_VIF*)** 1961–2011 Included Observations: 51 | | |
|---|---|---|---|
| $\overline{P}(i,t)j$ | **Coefficient Variance** | **Uncentered VIF** | **Centered VIF** |
| C | 20,915.97 | 70.89 | NA * |
| Ahome | 0.01 | 6.29 | 1.30 |
| Bocatoma | 0.05 | 37.89 | 3.67 |
| Choix | 0.02 | 37.15 | 3.10 |
| El Carrizo | 0.04 | 22.86 | 2.12 |
| El Fuerte | 0.06 | 73.86 | 4.46 |
| Higuera Zaragoza | 0.06 | 22.12 | 3.41 |
| Huites | 0.02 | 41.50 | 2.59 |
| Las Estacas | 0.07 | 46.83 | 2.76 |
| Los Mochis | 0.08 | 36.26 | 5.72 |
| Mochicahui | 0.05 | 13.56 | 2.09 |
| Presa Josefa Ortiz | 0.07 | 61.76 | 3.60 |
| Presa Miguel Hidalgo | 0.06 | 79.19 | 4.57 |
| Ruiz Cortínez | 0.05 | 28.99 | 3.28 |
| Topolobampo | 0.03 | 14.63 | 2.49 |
| Yecorato | 0.03 | 61.16 | 1.61 |

* Not apply, due to is one constant that not result of VIF calculation and is used for adjust the VIF results obtained with the aim to detect possible multicollinearity among predictor variables in the regression model of $\overline{P}(i,\ t)j$.

**Table 5.** The values obtained from *t_stat-DW*, ranging from 1.85 to 2.15, and the *t_stat-DFA* used to assess seasonality in the 16 historical datasets. *I*(1) indicated non-seasonality, while its absence indicated seasonality.

| | **Augmented Dickey–Fuller Test Statistic** Test on Null Hypothesis: Variable Has a Unit Root Exogenous: Constant, Linear Trend Lag Length: 0 (Automatic—Based on SIC, maxlag = 10) Sample: 1961–2011                        Included Observations: 51 | | | |
|---|---|---|---|---|
| | | **5% Test Critical Value = −3.5** | | |
| **Number** | **Variable** | **t_stat_DW *** | **t_stat_DFA *** | **p (Value) *** |
| 1 | Ahome | 2.02 | −6.25 | 0.01 |
| 2 | Bocatoma | 1.99 | −7 | 0.01 |
| 3 | Choix | 1.95 | −6.06 | 0.01 |
| 4 | El Carrizo | 1.97 | −6.27 | 0.01 |
| 5 | El Fuerte | 2 | 5.82 | 0.01 |
| 6 | Higuera Zaragoza | 1.96 | −7.005 | 0.04 |
| 7 | Huites | 1.96 | −7.07 | 0.00 |
| 8 | Las Estacas | 2.12 | −5.02 | 0.00 |
| 9 | Los Mochis | 1.98 | −5.93 | 0.00 |
| 10 | Mochicahui | 2.03 | −4.4 | 0.00 |
| 11 | Ocoroni | 1.83 | −6.02 | 0.00 |
| 12 | Presa Josefa Ortiz | 2.12 | −4.94 | 0.00 |
| 13 | Presa Miguel Hidalgo | 1.97 | −6.99 | 0.00 |
| 14 | Ruiz Cortínez | 1.95 | −5.51 | 0.00 |
| 15 | Topolobampo | 2.01 | −6.84 | 0.00 |
| 16 | Yecorato | 1.92 | −6.29 | 0.02 |

* *t_stat_DW* = Durbin–Watson Test, ** *t_stat_DFA* = Augmented Dickey–Fuller Test, and *** *p (value)* = Value Probability with respect to α = 0.05.

Table 5 shows the results of *t_Stat_DFA,* obtained by applying this test in its simple original manifestation and including those that were corrected via *STL_Decomposition,* and the *St* themselves, presented free of any process and in their seasonal form—that is, all tests were performed considering the first difference of $\overline{P}(i, t)j$ as a dependent variable. The *Sts* showed different positive or negative trends with respect to time, that is, without a tendency defined by a specific value around which each cloud of values was distributed. It was considered reasonable in the face of this behavior to perform the I of *t_Stat_DFA* without the existence of a slope. The analysis also considered the nature of positive values of the measurement with values above zero ($\overline{P}(i, t)j > 0$) in such a way that they did not manifest interception with the axis of the direction $x_i$. This behavior in the seasonal form of the *St* allowed us to perform *t_Stat_DFA* analysis in levels without tendency and without intercept and to omit the contrasts of *Stat_DFA* in first and second differentiation. Within this test for contrasts, we aimed to determine the possible existence of an *AC* in the information through the use of a Lag = 10 selected automatically in Eview 12.0 to compensate for this condition.

Before proceeding to accept or reject the $H_o$ hypothesis of seasonality using the contrast of the results of *t_stat_DFA* and the critical significance level of 5% equivalent to −3.5, it was first verified whether the value of *t_stat_DW* was included among the established limits of the range $R$ = [1.85, 2.15]. Consequently, ion limits were established. Then, with all certainty, we affirmed that an *AC* was not present in the model. It also indicated that the algorithms applied for the detection of non-seasonality together with the adjustment criteria had the expected results.

The datasets of 16 *Sts* are results based on *t_stat_DW, t_stat_DFA,* and *p_values* and they indicate the absence of *AC* within these models. Parallelly, the *t_stat_DFA* values, with respect to the critical level of significance of −3.5, also confirm the rejection of the $H_o$ hypothesis for seasonality in all 16 *St* datasets, as all *t_stat_DFA* values fall within the rejection zone (beyond the critical level of significance of −3.5). The results suggest that the *St* datasets, along with their autoregressive representations, do not display tendencies towards or equal to *I*(1), affirming their stochastic behavior with seasonal tendencies.

Additionally, the *p_value* variations ranging from 0.0001 to 0.004 further support the verification of seasonality within the *St* datasets. These contrasts provided high confidence levels in rejecting the $H_o$ hypothesis, with the errors in rejecting $H_o$ being lower than the accepted error level of 5%.

With the absence of *AC* established in the models, the nominal values of the 16 *Sts* exhibited consistent behaviors around specific values of $P = C$ and $\sigma^2 = K$. This assurance of stochastic, stationary, constant, and stable behaviors within these datasets facilitated the application of processes and algorithms, culminating in the determination of $P_T(i, t)j$ from 1961 to 2011 within each meteorological station. The spatial variation mentioned, shown in part A of Figure 6, represents the average sum of the 51 nominee values for each dataset. The datasets were transformed to the frequency domain to derive $/\overline{P}(r, t)_\theta/$ and were further analyzed through radial integration to ascertain $/\overline{P}(r)/$. The statistical results obtained before transforming $P_T(i, t)j$ into frequency domains and applying similar tools used for *AC* and seasonality verification on the 16 *Sts* are depicted in Table 5, indicating the rejection of the $H_o$ hypothesis of seasonality and affirming the presence of seasonality in $P_T(i, t)j$.

The final verification of the seasonality condition in $\overline{P}_T(i, t)j$ ensured that only stationary processes were employed throughout the study period, offering certainty regarding the spatial variability behavior and the gravity zoning effects caused by variable rain power in the study area, especially during the crucial initial 30 min where the most significant rainfall magnitudes occur, impacting the internal soil structure.

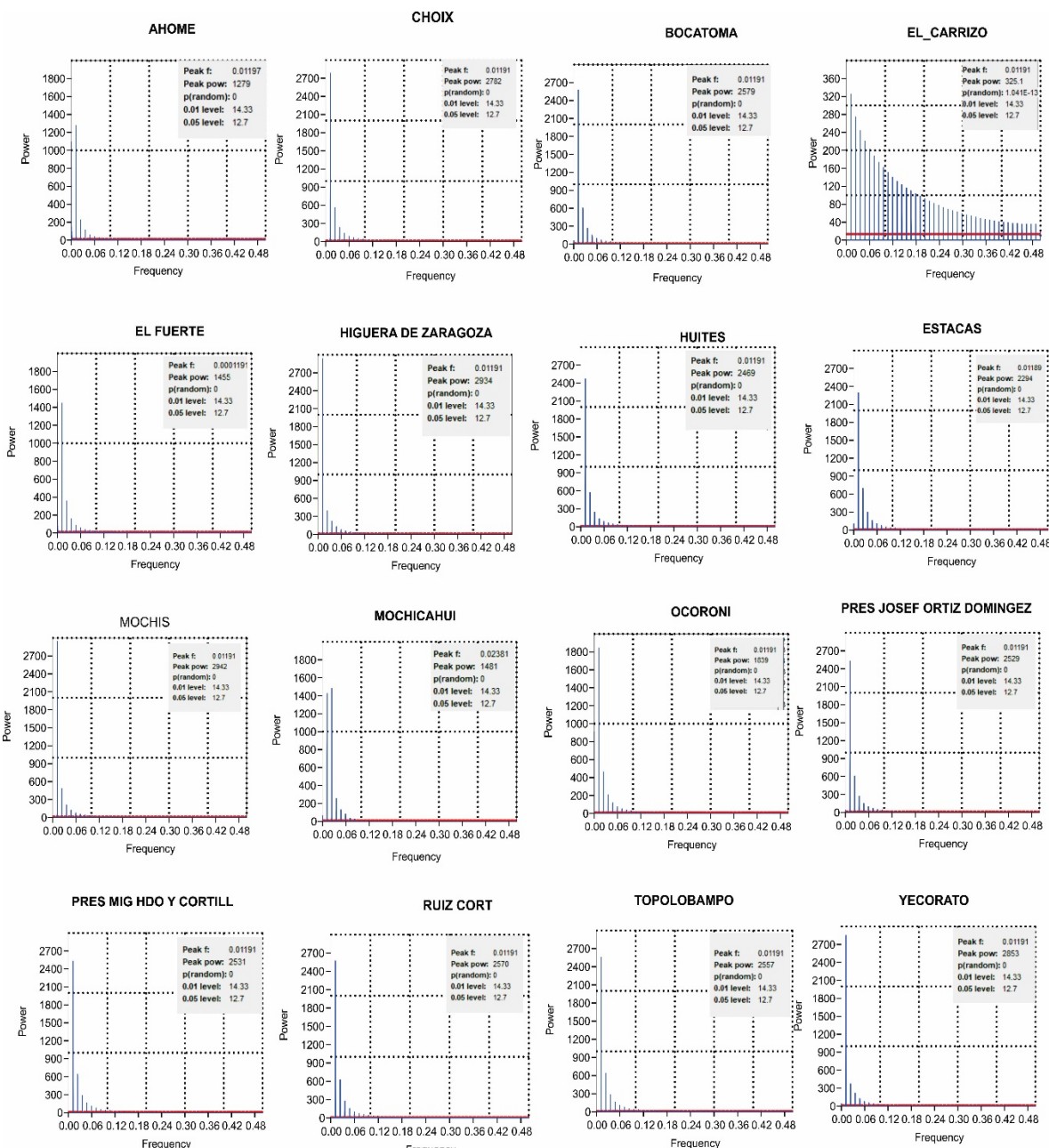

**Figure 6.** Radial potential spectrum depicting 16 historical stationary series of average annual total precipitation (mm), showcasing the magnitudes of initial impulses associated with moments of peak rainfall risk within the "*Río Fuerte*" and "*Río Sinaloa*" hydrographic and agricultural basins. The red line represents the random power (P(random)), and indicates the amount of noise present in the signal for each spectrum. When the red is intense, it corresponds to a high level of noise in the radial spectrum power.

The results of *t-stat_DFA* analysis for $\overline{P}_T(i, t)j$ are shown in Table 6 and they reveal that *t_stat_DW* = 1.99, which is within the range previously set as *R* = [1.85, 2.15], indicating the absence of *AC* in the model. Similarly, *t_stat_DFA* = −5.69 compared with *NC5%* = −3.06 signifies the acceptance zone, suggesting an absence of *I(1)* in the characteristic equation for both the original data and its first autoregressive representation. Consequently, $\overline{P}_T(i, t)j$ was determined to be a stationary stochastic process represented by $\overline{P}_T(i, t)j$ =504.53 mm and $\sigma^2 = 18,052.72$ mm in variance.

**Table 6.** Statistic Durvin–Watson test (*t_stat_DW*) and Augmented Dickey–Fuller test (*t_stat_DFA*) showing, respectively, the absence of autocorrelation and the presence of seasonality in the historical series of total annual average precipitation in the coastal plain and mountain zone of northwestern Sinaloa, Mexico.

| Augmented Dickey–Fuller Unit Root Test on PRECIP Null Hypothesis: PRECIP Has a Unit Root Exogenous: Constant, Linear Trend Lag Length: 0 (Automatic—Based on SIC, maxlag = 10) | | | |
|---|---|---|---|
| | | **t-Statistic** | ***p* (value)** |
| **Augmented Dickey—Fuller Test Statistic** | | **−5.69** | **0.01** |
| Test critical values: | 1% level | −4.15 | |
| | 5% level | −3.50 | |
| | 10% level | −3.18 | |

Dependent Variable: D(PRECIP)
Method: Least Squares
Sample (adjusted): 1962–2011
Included observations: 50 after adjustments

| **Variable** | **Coefficient** | **Std. Error** | **t-Statistic** | **Prob.** |
|---|---|---|---|---|
| PRECIP(-1) | −0.81 | 0.14 | −5.69 | 0 |
| C | 390.14 | 73.16 | 5.33 | 0 |
| @TREND("1961") | −0.02 | 0.88 | −0.02 | 0.98 |
| R-squared | 0.40 | | Mean dependent var. | −0.10 |
| Adjusted R-squared | 0.38 | | S.D. dependent var. | 115.25 |
| S.E. of regression | 90.52 | | Akaike info criterion | 11.90 |
| Sum squared resid | 38,517 | | Schwarz criterion | 12.02 |
| Log likelihood | −294.68 | | Hannan–Quinn criteria | 11.95 |
| F-statistic | 16.21 | | Durbin–Watson stat | 1.99 |
| Prob(F-statistic) | 0 | | | |

Via post-transformation of domains $/\overline{P}(r, t)_\theta/$ using *FFT*, the real shape of $/\overline{P}(r)/$ in the study area was understood, along with the intensity of the initial impulses representing the spectral power of rain in the different regions.

The spatial variability of the magnitude of the first impulses, represented in part B of Figure 6, illustrated a non-random pattern. Figure 6 portrays the graphical representation of $/\overline{P}(r)/$ along with the Neperian logarithms of $/\overline{P}(r, t)_\theta/$ in relation to the spatial frequency or wavenumber. The behavior of $/\overline{P}(r)/$ curves primary displays an exponential fall, with distinguishable linear relations between frequency ranges and their respective amplitudes. The exponential decay towards linear trends intensifies with increasing wavenumbers and potentially correlates with events of lower intensity originating from local sources.

Figure 6 presents a graphical representation of cycles per unit of distance, illustrating $/\overline{P}(r)/$ in terms of frequency and showcasing the magnitude of initial impulses, providing insights into the spectral power of rain.

These visualizations facilitate the comprehension of average total rainfall intensity at each meteorological station in relation to the logarithm of $[/\overline{P}(r)/]/4\,\pi$ and highlight distinct patterns of $/\overline{P}(r)/$ and its exponential decay across different stations. Moreover, the analysis of first impulses reveal unique magnitudes and frequency variations, with a notable concentration at 0.06 cycles/mm. Each meteorological station exhibits distinct behaviors in terms of rainfall intensity and frequency decay. Notably, the mesh of meteorological stations displays atypical power density behavior, indicating sequential and gradual changes in intensity related to local phenomena of lower intensity within its coverage area.

Figure 7, part A, illustrates the spatial variability of $\overline{P}_T(i, t)_j)$, revealing low accumulated historical rainfall at the Las Estacas meteorological station. This water deficiency poses a risk of water stress in this agriculturally significant area. However, for the same zone, despite the low accumulation, the rain intensity during the initial minutes suggests intermediate values, indicating potentially less damage to soil compared to "The Agricultural Heart of Mexico", which experiences a more substantial absence of rainfall. Additionally,

the El Carrizo meteorological station exhibits a gradual exponential decrease in rain frequency, which may affect surrounding regions with higher rainfall, potentially diminishing their influence.

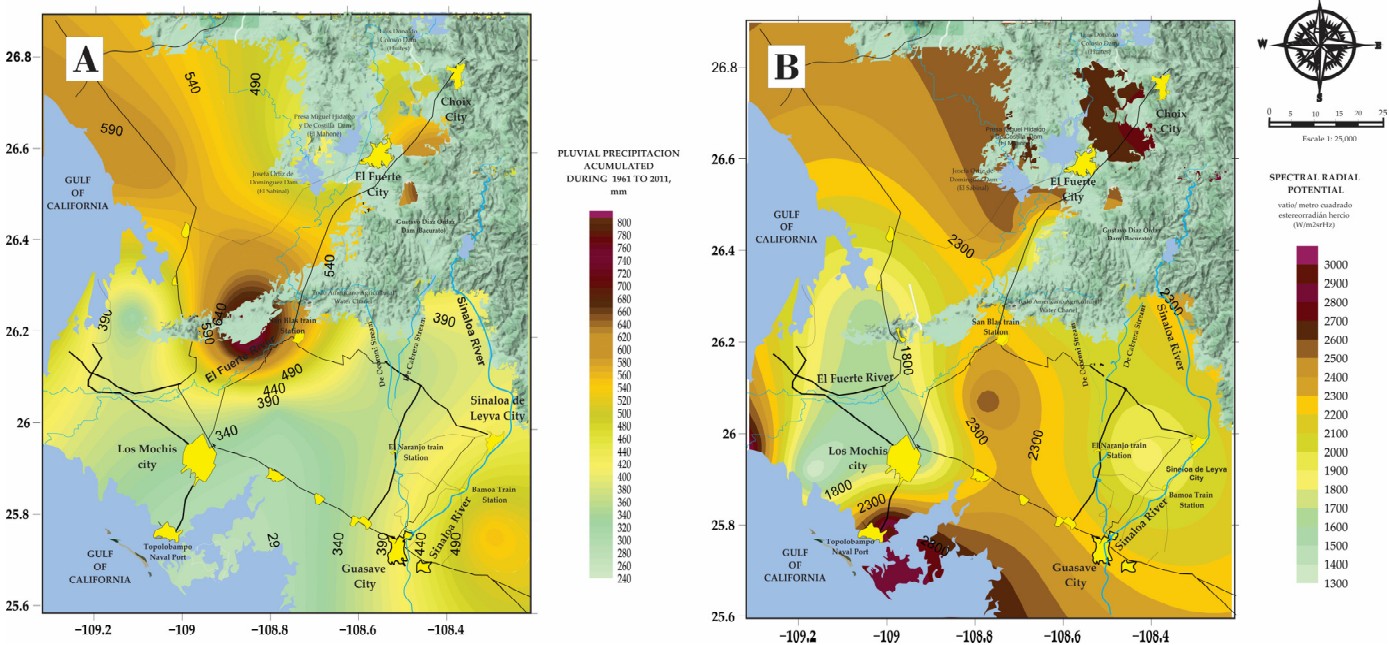

**Figure 7.** (**A**) Spatial variability of accumulated average annual total of precipitation ($\overline{P}_T(i, t)_j$) in mm: equipotential curves (620–800 mm). (**B**) Magnitude variations in initial impulses showing severe moments of soil degradation due to rainfall.

In the northern and central zones, there is noticeable variation in rainfall intensity, with maximum values ranging from 620 to 800 mm. Conversely, areas near the Sinaloa river mouth and the Sea of Cortez exhibit minimal rainfall magnitudes, ranging from 240 to 360 mm. While these minimal magnitudes support agricultural activities, they can also lead to water deficits and impact soil conditions due to high evapotranspiration rates compared to local water from precipitation and groundwater flow.

This spatial variation in rainfall intensity and its initial impulses $/\overline{P}(r)/$ across maps in Figure 7 are attributed to variable microclimates affecting meteorological factors, generating convective precipitation typical of warm latitudes. Variability in atmospheric pressure, temperature, and humidity within these microclimates contributes to the diverse patterns observed during the initial moments of rainfall, crucial for understanding how rain impacts the granulometric structure of the soil.

Analysis underscores the varying consequences and severity of rain's effects on soil structure, particularly during the initial moments of rainfall, which pose a risk to soil integrity. Different microclimates across the study area significantly impact the severity of consequences resulting from rainfall power, particularly in the first 7 to 30 min, crucial for assessing soil degradation risks.

The risk is more pronounced in mountainous areas and the central zone parallel to the Sea of Cortez. It highlights the potential for accelerated degradation of agricultural soils due to water deficit and structural changes caused by rainfall. These severe consequences are not limited to "*The Agricultural Heart of Mexico*" and also extend along the coastal regions, demanding urgent attention for soil restoration, considering the implications of climate change and globalization demands.

Figure 8 illustrates the severity levels of consequences from the initial minutes of $/\overline{P}(r)/$. Over time, the highest severity levels extend across coastal areas, potentially accelerating land degradation. Three categories of severity with variable impacts on soil were identified:

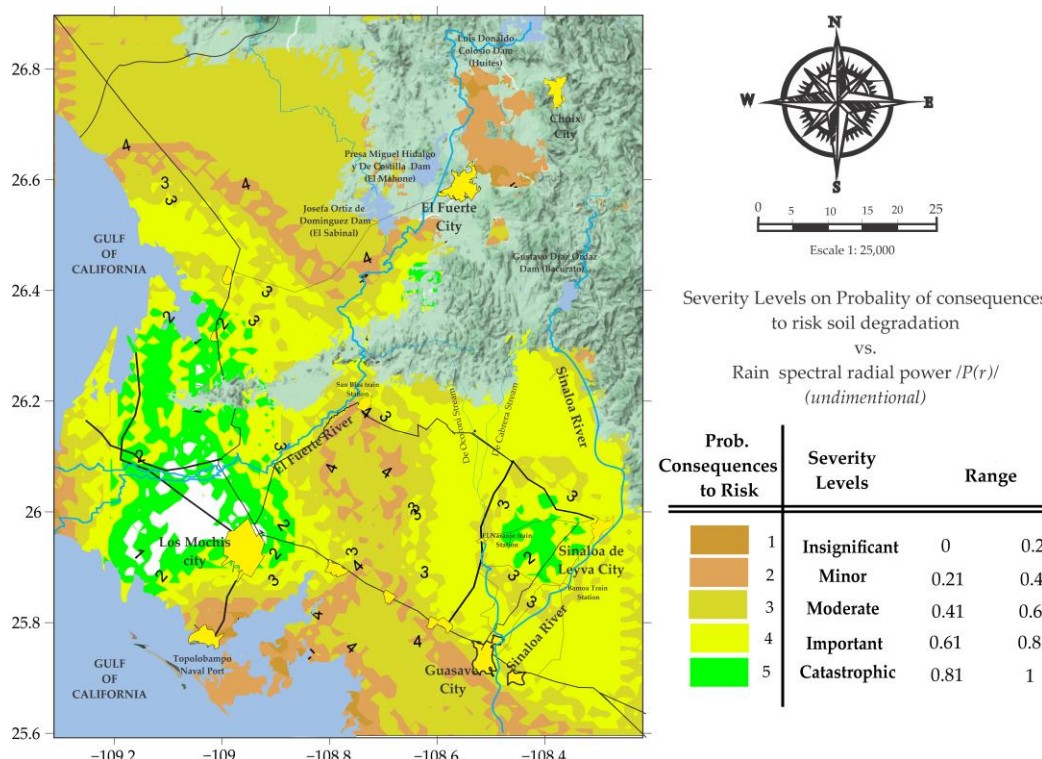

**Figure 8.** The classification of severity levels of consequences (dimensionless) on soil, resulting from the effects of historical accumulated rainfall, situation that could accelerate soil degradation.

**Moderate Consequences**: these areas demand immediate conservation actions to mitigate the impacts, as the consequences will persist and require significant effort for restoration.

**Important Consequences**: This severity category requires detailed studies to understand the current soil conditions. Immediate actions are needed to stabilize the soil structure and regulate its properties, considering porosity, water retention capacity, and organic matter development.

**Catastrophic Effects**: These areas have sustained severe damage and exploitation without proper consideration. They are currently unsuitable for sustainable agriculture and necessitate exhaustive recovery efforts for long-term restoration due to their difficult recovery state.

The classification of severity helps identify regions that need urgent attention for conservation, areas requiring comprehensive studies for stabilization, and those that demand long-term recovery strategies due to their current unsustainable state.

## 4. Discussion

The information gathered constitutes a comprehensive database and methodology that employs a suite of contemporary statistical tools. These tools are applied to generate crucial insights for the development of management plans aimed at risk mitigation based on the severity of consequences resulting from rainfall, a pivotal factor influencing soil structural degradation. This information also holds potential utility for future research endeavors, territorial planning initiatives, and various governmental or private projects focused on conserving agricultural soil in the region.

Presently, the region, comprised of the key valleys in northwestern Mexico, lacks adequate recognition in the international agricultural market. The environmental value of its goods and services is not duly acknowledged, leading to a disparity in fair representation. This oversight is particularly concerning given the historical damage inflicted on the soil, directly impacting productivity across the agricultural cycle from planting to distribution.

To address these challenges, the region requires immediate optimization. Neglecting the true cost of agricultural products within the value chain leads to market failure, which negatively impacts overall costs and subsequently affects the economy and welfare of the population in Mexico [46]. The adverse effects of climate change on various global agricultural systems have been exacerbated by recent data released by the World Meteorological Organization (*WMO*) in the document "State of the Global Climate 2023" on 19 March 2024. According to this report, 2023 was the warmest year on record, with an average surface temperature increase of approximately $1.45 \pm 0.12$ °C compared to pre-industrial levels (1850–1900) [47]. Following this trend observed in 2023, the historical changes in annual average global surface temperatures reach a total average increase of 1.5 °C, surpassing the predicted 1.2 °C estimated for 2025. This further underscores the urgency of addressing environmental issues in Mexico [48].

Changes in environmental components due to climate change in the region's valleys, as indicated by the historical $\overline{P}(i, \ t)j$, may contribute to accelerated soil cultural degradation. The situation described could jeopardize "*The Agricultural Heart of Mexico*", potentially affecting crop physiological processes, growth, and overall production. To address these challenges, the region requires immediate optimization of both surface and underground water resources. Water deficits, aggravated by the arid and semi-arid climate and climatic instabilities, pose a significant threat to sustainable agriculture [49]. The findings highlight a water deficit persisting for decades, impacting soil structural arrangements and emphasizing the need for continued research.

To ensure sustainable agriculture in the region, a shift towards water-efficient crops is imperative. Studies focusing on crops with a balance in water management, emphasizing productivity efficiency, are crucial for adapting to the limited water environment [48,49]. The detected changes in the soil structure demand a commitment to repairing the damage caused and identifying the associated risks and consequences, categorizing them based on severity.

Quantifying *GHGs* associated with food production is essential for addressing climate change impacts and responses to climate change, according at its opportunities and challenges [50]. The valuation of environmental assets, including soil resources, must become a priority in sustainable development, attributing fair environmental values based on soil characteristics. Agricultural sustainability necessitates a restructuring of natural resource management and parallel routes to international trade openness.

In this era of modern globalization, Mexican agriculture must strive for sustainability. Achieving this goal involves restructuring natural resource management and adopting parallel strategies for international trade openness. The impacts of human and natural activities altering the particle size structure of the soil must be considered in management plans for the correct and sustainable utilization of natural resources.

Assigning fair environmental values to agricultural assets, with indicators designed based on land use, *GHGs*, risk, and current vulnerability, is crucial. This is particularly relevant in the case of rainfall, a phenomenon sensitive to climate change. The information generated in this study proves useful in developing indicators that provide a fair value to agricultural assets, especially in "*The Agricultural Heart of Mexico*".

For the case study, rain risk in the soil was utilized as a spectral metrology scheme to analyze energy variations concerning the historical variations of each *St*, aiming to determine the severity levels of damages on the soil. It is important to note that interpretations can be complex and may potentially lead to contradictory or confusing conclusions due to the inherent presence of a minimum percentage of randomness in the data that has not been entirely eliminated. Addressing these issues requires the correction of randomness in the data series, specifically sporadic impulses or signals inserted irregularly. This study, focusing on the seasonality of the series, facilitated the observation of real and cyclical behaviors, enabling meaningful contrasts between signals. This approach is crucial for understanding the severity of consequences resulting from rainfall and its impact on the structural integrity of agricultural land crucial for the economy of Mexico.

Regardless of the specific methodology employed, it is essential to correct and adjust time series data for seasonality before analysis. Failure to do so can lead to misinterpretations and false spatiotemporal behaviors. To address these challenges, before making any important decision that depend on the signals with variations in time, three types of harmonics have to present—original, corrected for calendar effects, and adjusted for seasonality—to eliminate randomness. Also, a methodological note should accompany each indicator to mitigate potential confusion among analysts.

Hence, is necessary to foster a statistical and econometric culture, especially in countries in development like Mexico. The lack of familiarity with econometrics in obtaining political, social, and economic indicators hinders accurate projections, often leading to responses influenced by regionalist traditions. Establishing a robust statistical culture is crucial for effective decision-making and policy formulation in search of development.

This finding emphasizes the need for sustainable agriculture in the agricultural valleys of Mexico. Assigning appropriate environmental values to agricultural assets is crucial for achieving agricultural sustainability. This involves restructuring natural resource management, quantifying damages, and acknowledging the environmental impact of climate change. Harmonizing statistical and econometric practices is essential to ensure accurate projections and avoid misinterpretations, contributing to the overall progress of countries like Mexico.

## 5. Conclusions

The findings of the study emphasize the pivotal role of precipitation intensity, as demonstrated using metrics such as $/\overline{P}(r,\,t)_\theta/$ and $/\overline{P}(r)/$, in shaping soil management practices within the examined area. This underscores the urgent need for collaborative efforts involving agricultural producers, local communities, academic researchers, and governmental institutions at municipal and state levels to effectively address this phenomenon. Identifying and tackling key factors that impede soil conservation in one of Mexico's vital food production regions is paramount. Specifically, strategies should be tailored to mitigate the impacts of climate change on agricultural soils, with a particular emphasis on alleviating water stress resulting from both excessive and deficient precipitation.

"Collaborative partnerships with governmental institutions are crucial for the development and implementation of effective local environmental policies"; specifically, collaborations with governmental bodies are pivotal for formulating and executing effective local environmental policies, where agriculture should not only be seen as a means of food production but also as an integral part of inclusive climate justice initiatives [51]. Climate justice has emerged as an effective strategy to address the inequalities and injustices stemming from climate change impacts on natural resources in various regions, not only in Mexico but also worldwide.

In the broader context of global sustainable development and amidst the competitive pressures of contemporary globalization, the enduring agricultural traditions within this region boasting dual cropping seasons (spring–summer and summer–autumn), and recent technological advancements, the "*Agricultural Heart of Mexico*" it has firmly established itself as a frontrunner in global food production. This underscores the urgent imperative to confront the challenges stemming from the degradation of agricultural land attributed to a myriad of environmental, political, economic, and cultural factors that contribute to the erosion of soil granulometric structure.

Addressing these challenges in regions where the repercussions of risks have proven to be calamitous is essential to ensure the continuity of agricultural productivity and to maintain an uninterrupted provision of nutritious food. Such efforts significantly contribute to global endeavors aimed at mitigating food insecurity in a world with a continuously expanding population. It is paramount to sustain this level of sustainability and competitiveness, not only for this pivotal agricultural hub but also for numerous other agricultural regions across Mexico.

There is a pressing need to revamp agricultural business strategies, shifting the perception of agriculture from mere food production to a pivotal element within inclusive climate justice initiatives. This paradigm shift ensures that farmers play a crucial role in bolstering both national food security and global food supply chains. The emergence of climate justice as an effective strategy to combat the inequalities and injustices stemming from climate change impacts in diverse regions further underscores the urgency of these reforms.

Efforts should be focused on conserving agricultural soils and addressing issues such as saline water intrusion, water stress, desertification, and erosion, particularly water erosion resulting from the diverse potency of rainfall in the area, which poses a significant threat to soil fertility and landscape integrity and impacts the economy and welfare of society.

Given the changing precipitation patterns attributed to climate change, the information gleaned from this study is invaluable for designing tailored soil management plans. These plans should prioritize measures to mitigate the adverse impacts of water erosion, considering the specific characteristics and associated risks of different soil types in the region. Unit root tests, autoregressive equations, and multivariate methods were also employed in the analysis, enhancing the robustness of the findings.

Since a significant portion of soils in the region comprise fine grains, it is advisable to conduct laboratory soil analyses to obtain indicators that can indirectly assess water retention based on soil internal granulometry. This process involves examining soil alterations at three Atterberg limits (contraction or consistency, plasticity, and liquid) to establish correlations between these limits and the soil's capacity for water adsorption [52]. Omitting these activities in the future could result in severe damage to the soil structure, rendering it unsuitable for agricultural purposes and diminishing the economic benefits derived from agricultural activities. Therefore, concerted efforts are needed to safeguard Mexico's agricultural resources and promote sustainable land management practices for the benefit of present and future generations [53,54].

The observed soil degradation in the valleys underscores the critical need to implement effective management practices to preserve this essential resource, as discussed in [55] (pp. 174–176), emphasizing that preserving soil biodiversity is necessary to maintain soil productivity. Additionally, this management is crucial to preserve the deeply intertwined cultural heritage associated with these invaluable soils. These lands have a rich agricultural history dating back to pre-Spanish conquest times, during which agriculture played a fundamental role in meeting the nutritional needs of indigenous communities in the region. Conserving these soils not only ensures sustainable agricultural practices but also honors and sustains the cultural legacy for future generations from these first Mesoamerican indigenous population.

**Author Contributions:** Conceptualization, M.N.C.; writing and preparation of original draft, M.L.d.G.T. and P.M.S.; methodology, M.N.C. and M.L.d.G.T.; formal analysis of present draft, J.M.M. and L.A.S.G.; data preparation, M.L.d.G.T. and O.L.C.; writing—review and editing, M.N.C.; L.A.S.G. and J.M.M.; project administration, M.L.d.G.T. All authors have read and agreed to the published version of the manuscript.

**Funding:** This research was funded by the Research and Postgraduate Secretary Office (SIP) of the National Polytechnic Institute (IPN), through annual calls for individual and multidisciplinary projects (No. 20220924, 20231294, and 20241198, as well as by the municipal governments of Sinaloa, located in northwest Mexico.

**Institutional Review Board Statement:** Not applicable.

**Informed Consent Statement:** Not applicable.

**Data Availability Statement:** The data presented in this study are available on request from the corresponding author. The data were not directly appended to this document, as the statistical methods employed, as well as the datasets utilized to derive the results, are comprehensively described in the manuscript.

**Acknowledgments:** The authors express gratitude to SIP of the IPN for their continuous financial support and their encouragements throughout all stages of this research, including fieldwork, office work, and resource and material management, among others. Additionally, we extend our acknowledgments to the editor and anonymous reviewers. Special appreciation is extended to individuals from CIIDIR departments of investigation or other research centers affiliated or not with SIP who provided critical comments, as their insights were instrumental in developing the arguments presented in this study. The SIP of the IPN and colleagues from the Departments of Environmental and Natural Resources, Aquaculture, and Biotechnology at CIIDIR-IPN Sinaloa Unit consent to this acknowledgment.

**Conflicts of Interest:** To mitigate any potential influence due to conflicts of interest, several measures have been implemented to ensure transparent and comprehensive disclosure throughout all phases of the present research. Complete transparency is ensured by disclosing the sources that provided funding for the development of this research. Additionally, a rigorous approach was taken in data collection and analysis, with explicit mention of the sources from which data was obtained. The methodology employed is based on tools that can be independently verified, thus remaining unaffected by external influences on the research. Collaboration with experts in the field was actively sought to provide diverse perspectives not tied to any particular interests but rather solely focused on an objective assessment of the developments and analyses conducted.

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
