# Peer review of "Rainfall Potential and Consequences on Structural Soil Degradation of the Most Important Agricultural Region of Mexico"

_atmosphere, doi:10.3390/atmos15050581_

Round 1

Reviewer 1 Report

Comments and Suggestions for Authors

Dear authors,

The work covers an important and interesting subject and contributes new insights to the literature. However, The methodology section of the manuscript is currently presented in a narrative style, spanning 5-6 pages without clear subdivisions. Also, the article appears to deviate from the main principles that the articles depend on, including a robust theoretical framework, methodological framework, and structured presentation of findings. The methodology, introduction, and results sections must be revised and developed. I recommend adding a flowchart to simplify the methodology. This paper seems limited in methodology and structure and needs more work. I believe it must be accepted with a major revision in its current form.

I list out some main concerns below, and then the comments for the lines.

Major comments:

·       The article appears to deviate from the main principles that the articles depend on, including a robust theoretical framework, methodological framework, and structured presentation of findings. Without these essential elements, the article's potential contribution to the field is limited.

·       The methodology section of the manuscript is currently presented in a narrative style, spanning 5-6 pages without clear subdivisions. To enhance the clarity and readability of this section, it is recommended to restructure it by introducing clear subsections. Each approach, equation, or method should be explained individually to provide a step-by-step understanding of the research methodology. This restructuring will not only simplify the reading process but also facilitate a more organized and systematic presentation of the research methods employed. Clear subsections will guide the reader through the methodology, ensuring that each component is distinctly explained and contributing to a more comprehensive understanding of the research design.

·       It is advisable to incorporate a flowchart in the methodology section to visually illustrate the step-by-step procedures, enhancing the overall comprehension of the research design.

·       Which spatial interpolation technique did you use in the study?

·       The introduction doesn't effectively set up the problem, and the flow of the content is somewhat disjointed. It requires substantial revision and enhancement. It currently contains considerable unnecessary information that does not contribute directly to establishing the context or highlighting the research gap. Additionally, several statements lack proper references, and the overall number of references is insufficient to support the presented information comprehensively. To improve the introduction, it is recommended to streamline the content, focusing on providing concise background information, clearly defining the research problem, and establishing the significance of the study within the existing literature. Moreover, ensuring that all statements are appropriately referenced will enhance the scholarly integrity of the introduction section.

·        The conclusion section should focus on summarizing the key findings and their implications, avoiding unnecessary repetition to enhance clarity and conciseness.

·       Kindly read the authors' guidelines. You have to meet the journal guidelines to improve the manuscript.

Minor comments:

Line 12: check it.

Lines 60 – 65: delete it or keep it with references.

Lines 104 – 113: I think there is no need for this paragraph.

Lines 114 – 116: Why did you separate the research objectives in one sentence and paragraph?

Figure 1: it is unreadable. Improve the quality of this figure. Also, what is the yellow area in the figure? Check the legend.

Figure 2: Figure 2.A has a lot of data that made the figure unreadable. Nothing can be read or noticed from this figure.

Line 638 – 687: revise these paragraphs.

Why did you use this font in the tables? Modify all tables.

Sincerely,

Comments on the Quality of English Language

Minor editing of English language required

Author Response

Dear reviewer of manuscript Atmosphere-2925268,

I begin this document by acknowledging you the importance of your work regarding the significant observations made to mentioned manuscript, which its I found very precise to improving this writing since the first time that these was received.

Below, I will to provide the context for each of the observations and the improvements done to the original manuscript, either in its entirety or specifically addressing the observations reviewed collectively by you:

  • Regarding to methodology section, which was presented in a narrative style in the original document, it has now been restructured objectively. Despite the complexity of the topic covered by the various methodologies used for analysis, there was no substantial reduction in content. However, following your recommendations, methodological segmentation was carried out to comprehend and describe the sequential methods used to establish seasonality in the time series data.
  • Similarly, in response to your recommendation to include a flowchart illustrating the research framework step by step, such a diagram has been included from the outset in the initial sections of the methodology. A brief description is provided within the manuscript to highlight the purpose of its inclusion, emphasizing that it is intended to provide readers with an organized and systematic understanding of the research methods employed. It is hoped that this approach meets your expectations by clearly guiding the reader through the methodology, ensuring that each component is clearly explained and contributes to a comprehensive understanding of the research design.
  • Regarding the interpolation technique used, a detailed description was provided regarding the testing of a set of kriging methods evaluated for differences in the percentage of explained variability and root-mean-square error found in cross-validation. Of the various interpolation techniques assessed (ordinary kriging, universal kriging, kriging with external drift, with individual variograms, and with combined variograms), the one utilizing combined variograms was selected for its comparable performance relative to other interpolators. This method allowed for the configuration of distributions representing the real total accumulated precipitation by rainfall in the study valley. The response of this kriging method in special with combined variograms regarding its spatial representation of meteorological dataset, had been used in previous research, and citations of some these works have been included.
  • The introduction part, it was rephrased to align with the content found in the original document, giving it a coherent flow and incorporating recommended information to highlight the research focus and to support the assertions made in each paragraph. Only the ideas contributed by myself in the research not were cited, while some paragraphs hat do not was contain citations were omitted because was considered that were out of context, not explained and not contributes to a comprehensive understanding of the research design. Is expected that these assertions, now support by references will give a solid scientific basis, meet the expected standards to ensure that statements are adequately referenced to enhance this section of the manuscript.
  • The conclusion part was also rewritten, focusing on the key findings and implications of effects accumulated by rainfall in light of current climate change. Several recent dates were included information of new reports of temperature global changes brought about by climate change, highlighting the provided by the World Meteorological Organization's in recent inform data release. Among other citations, these reinforce the key findings and implications regarding soil degradation in this important agricultural valley of our country.

  • Tables, figures and the flowchart according to the guidelines provided to authors by this journal both were done.

Regarding minor comments, responses to each of these are provided below:

  • Line 12: Revised. The number three was corrected as it appeared repeatedly due to institutional duplication.
  • Lines 60-65: Deleted or maintained with references. Salinas and others were removed from the manuscript as the statement lacked context and references, and was deemed irrelevant. Although these methodologies are applied to the study of other systems, they are not applicable in this context.
  • Lines 104-113: This paragraph was deemed unnecessary and has been removed.
  • Lines 114-116: Objectives were rewritten together with justification for addressing the problem, ensuring coherence and clarity.
  • Figure 1: Improved the quality and added a legend to clarify the yellow area.
  • Figure 2: Split into two figures to enhance readability and avoid overcrowding.
  • Lines 638-687: These paragraphs were reviewed and rewritten.
  • Font usage in tables: Adjusted to adhere to journal guidelines.

Lastly, regarding your comments on the English language, I appreciate your understanding of the minor edits required in this language, which is not my native tongue.

Again, permit me to say you, much thank you for your thorough review and valuable input to this manuscript and receive a warm greeting and appreciation for your comments and those of other reviewers, to the significant modifications and improvements made to the manuscript. We remain at your disposal for any further inquiries.

Sincerely,

Mariano Norzagaray Campos and Authors

Reviewer 2 Report

Comments and Suggestions for Authors

The study investigates how historical rainfall variations impact soil degradation in Mexico's northwest region. Analyzing sixteen rainfall series, it finds severe soil damage correlated with unpredictable rainfall patterns attributed to climate change. Emphasizing the need for tailored numerical processes, the study highlights the urgency of understanding soil responses for sustainable management practices. The manuscript is well written but still lacks clarity in highlighting its novelty, which is crucial for distinguishing its contribution to the field. Furthermore following points can be considered to improve the manuscript-

1. Methods like GCE Analysis rely on visual interpretation of geometric shapes, introducing subjectivity and potential bias in identifying autocorrelation patterns. This could lead to inconsistent results depending on the observer's interpretation.

2. While the manuscript mentions various software tools used for data analysis and visualization, it lacks detailed explanations of how these tools were applied and their specific contributions to the methodology. Providing more clarity on the role of each software tool would enhance transparency and reproducibility.

3. The use of kriging interpolation for spatial variation analysis may introduce uncertainties, especially if the interpolation parameters are not well justified or validated. A thorough explanation of the interpolation methodology and its limitations would strengthen the validity of spatial analysis results.

4. The severity scale for consequences of rainfall patterns introduces subjective categorization and may lack universal applicability. Justification for the chosen categories and validation against real-world data or expert opinions would improve the credibility of the severity assessment. 

5. The methodology heavily relies on statistical parameters and tests to verify the presence of autocorrelation and stationarity. However, these tests may not capture all nuances of the data, and additional validation through alternative methods or sensitivity analyses could provide more robust results.

Comments on the Quality of English Language

The quality of English in the manuscript appears to be satisfactory overall. The language is clear and understandable, with technical terms appropriately used. However, there are a few instances of awkward phrasing and grammatical errors that could be addressed for improved readability and coherence. Overall, minor revisions may be needed to enhance the manuscript's clarity and fluency.

Author Response

Dear Reviewer of Manuscript Atmosphere-2925268,

Allow me to begin by expressing my appreciation for your diligent work on the mentioned manuscript. The detailed observations you provided have been thoroughly considered and warmly received as invaluable insights for enhancing this manuscript. Below, I outline the actions taken in response to your insightful recommendations:

  • Regarding the tools and software utilized in data analysis and visualization, we acknowledge the necessity of providing a detailed explanation of their application and specific contribution to the methodology. To enhance transparency and reproducibility, we have objectively restructured the methodology, addressing the complexity of the methodologies employed in the research. The original version was presented in a narrative style; it has now been restructured objectively. Despite the complexity of the topic covered by the various methodologies used for analysis, there was no substantial reduction in content. However, following your recommendations, despite the section's length, we have achieved a clearer and more sequential description of the methods used to establish seasonality and autocorrelation in the analyzed time series. With this sequential methodological segmentation, it is expected that the methods used to establish seasonality in the time series data are understood and described.
  •  Additionally, we have included a flowchart in the initial sections of the methodological framework to provide readers with an organized and systematic understanding of the research methods employed. A brief description is provided within the manuscript to highlight the purpose of its initial inclusion, emphasizing that it aims to provide readers with an organized and systematic understanding of the research methods employed. This flowchart illustrating the research framework step by step has also been segmented, and each sector has been elaborated regarding the sequential use of the tools. It is hoped that this approach meets your expectations by clearly guiding the reader through the methodology, ensuring that each component is clearly explained and contributes to a comprehensive understanding of the research design.
  • Regarding the interpolation technique used, a detailed description was provided regarding the testing of a set of kriging methods evaluated for differences in the percentage of explained variability and root-mean-square error found in cross-validation. Of the various interpolation techniques assessed (ordinary kriging, universal kriging, kriging with external drift, with individual variograms, and with combined variograms), the one utilizing combined variograms was selected for its comparable performance relative to other interpolators. This method allowed for the configuration of distributions representing the real total accumulated precipitation by rainfall in the study valley. The response of this kriging with combined variograms regarding its spatial representation of precipitation had been used in previous research, and citations of these works have been included.
  • The introductory part has been reformulated to align with the content found in the original document, providing a coherent flow and incorporating recommended information to highlight the research focus and support the assertions made in each paragraph. Only the ideas contributed by myself in the research were cited, while some paragraphs that did not contain citations were omitted because they were considered out of context, unexplained, and did not contribute to a comprehensive understanding of the research design. It is expected that these assertions, now supported by references, will meet the expected standards to ensure that statements are adequately referenced, thereby enhancing scientific agreement on this section of the manuscript.
  • The conclusion was also rewritten, focusing on the key findings and implications in light of current climate change. Several recent graphics were included to compare the temperature changes brought about by climate change, highlighting alarming considerations provided by the World Meteorological Organization's recent data release. Among other citations, these reinforce the key findings and implications regarding soil degradation in this important agricultural valley of our country.

  • Tables, figures, and the flowchart have been improved throughout the manuscript according to the guidelines provided for authors of the journal.

  • In addition, adjustments have been made regarding the subjective severity classifications for the consequences of accumulated rainfall patterns, including a brief description supported by expert opinions from CONAGUA to present them in the results in a specific manner that adapts to the severity response of the consequences of accumulated rainfall in the study area.

Lastly, regarding your comments on the English language, I appreciate your understanding of the minor edits required in this language, which is not my native tongue.

Receive a warm greeting and appreciation for your comments and those of other reviewers, to the significant modifications and improvements made to the manuscript. We remain at your disposal for any further inquiries.

Sincerely,

Mariano Norzagaray Campos and Authors

Round 2

Reviewer 1 Report

Comments and Suggestions for Authors

Dear authors,

I would like to thank the authors for their efforts and work. They addressed most of my concerns and questions. The revisions made to the manuscript have significantly enhanced its quality and clarity. However, I have specific points that require further clarification or adjustment. I kindly request that you consider addressing these points in your final revisions.

·       The numbers in the tables do not require multiple decimals; 2 or 3 decimals would be sufficient.

·       The font and font size within the manuscript must be consistent across all sections, tables, figures, etc.

·       The quality of Figures 1, 4, and 5 must be improved.

Sincerely  

Comments on the Quality of English Language

Minor editing of English language required

Author Response

Dear Reviewer

      First and foremost, I want to express my sincere gratitude for the accurate work of your corrections and the improvements they brought to the manuscript during your reviews. Your dedication and attention to detail have been invaluable in elevating the quality and clarity of the work, and I am genuinely thankful for that.

      We have reviewed and addressed the concerns raised by the reviewer in our scientific manuscript. We sincerely appreciate your efforts and work in this process.

We have made the necessary corrections, including reducing decimals in the tables and standardizing the font type throughout the manuscript, as requested. However, I would like to briefly discuss an issue related to the figures.

    We understand the importance of maintaining consistency in the font type throughout the manuscript, including the figures. Therefore, we want to emphasize that the same font type was maintained in all figures. However, upon careful review of some of the figures, we realized that in certain cases, maintaining the same font size could result in the loss of clarity or lack of emphasis on important aspects within the figures.

Therefore, we made the decision to adjust the font size in some instances to properly highlight certain relevant aspects in the figures. We believe that this adjustment improves the overall understanding of the presented information.

We are open to any further suggestions you may have on this matter, and we are willing to make the necessary changes to ensure the quality and coherence of the manuscript as a whole.

Thank you again for your collaboration and support in this process.

Sincerely,

Mariano Norzagaray Campos and authors